# Polynomial, trigonometric, and tropical activations

**Ismail Khalfaoui-Hassani & Stefan Kesselheim**
Jülich Supercomputing Centre
Forschungszentrum Jülich
Jülich, Germany
{i.khalfaoui,s.kesselheim}@fz-juelich.de

## Abstract

Which functions can be used as activations in deep neural networks? This article explores families of functions based on orthonormal bases, including the Hermite polynomial basis and the Fourier trigonometric basis, as well as a basis resulting from the tropicalization of a polynomial basis. Our study shows that, through simple variance-preserving initialization and without additional clamping mechanisms, these activations can successfully be used to train deep models, such as GPT-2 for next-token prediction on OpenWebText and ConvNeXt for image classification on ImageNet. Our work addresses the issue of exploding and vanishing activations and gradients, particularly prevalent with polynomial activations, and opens the door for improving the efficiency of large-scale learning tasks. Furthermore, our approach provides insight into the structure of neural networks, revealing that networks with polynomial activations can be interpreted as multivariate polynomial mappings. Finally, using Hermite interpolation, we show that our activations can closely approximate classical ones in pre-trained models by matching both the function and its derivative, making them especially useful for fine-tuning tasks. These activations are available in the torchortho[1] library.

## 1 Introduction

Modern deep learning is largely built upon the Multi-Layer Perceptron (MLP) McCulloch & Pitts (1943); Rosenblatt (1958) and the gradient backpropagation algorithm Rumelhart et al. (1986). The MLP can be described as a combination of a multiplication by a matrix of learnable weights and the application of a nonlinear activation function. Gradient backpropagation, on the other hand, relies on the chain rule to compute partial derivatives necessary for optimizing weights through gradient descent. In a deep neural network, *preserving variance across layers* is critical to ensure stable training dynamics. Glorot & Bengio (2010) and He et al. (2015) were the first to consider a variance-preserving analysis for deep neural networks.

The analysis shown in He et al. (2015) could be stated as *the output signal of each MLP block should have the same variance as the input signal*. And since learning is performed with backpropagation, this same rule should apply to the gradients as well, meaning that *the variance of the gradient of the input should also be equal to the variance of the gradient of the output of the MLP*.

In this manner, He et al. (2015) demonstrated the methodology for initializing the weights of a deep neural network, thereby attaining performance on ImageNet classification that exceeds that of humans. This process entails the calculation of the ratio between the variance pre- and post-activation, called forward gain, as well as the ratio of variance with respect to the derivative of the activation, called backward gain. Remarkably, for the ReLU function, both forward and backward gains are equal to 2.

Recently, Yang & Wang (2025) employed the same principle to train learnable rational activations. However, they encountered a challenge: the second-order moment has no closed formulation in the case of rational fractions. The authors' solution for ensuring the convergence of such rational activation networks consisted in initializing them by fitting the polynomial coefficients to a classical

---

[1]https://github.com/K-H-Ismail/torchortho

activation such as ReLU or SiLU Ramachandran et al. (2018); Elfwing et al. (2018). Here, we propose a solution to the aforementioned problem by employing orthogonal basis functions (Fig. 4), specifically polynomial and trigonometric functions. Orthogonal basis functions in a chosen $L^2$ space, as will be elucidated in the subsequent sections, facilitate the calculation of the second-order moment integral, thereby yielding a closed and straightforward formula. Additionally, we demonstrate that rational functions are unnecessary, asserting that polynomial activation functions are sufficient.

More generally, the convergence of polynomial networks shown in this work proves that deep neural networks can be seen as multivariate polynomial mappings. Indeed, the successive layers of a feed-forward network activated by a polynomial activation can be seen as a composition of weighted sums of multivariate polynomials, ultimately resulting in a polynomial mapping. A parallel representation was made by Zhang et al. (2018) for ReLU-activated networks, demonstrating that they are tropical rational mappings. In a later section, we also explore tropical polynomial functions as activation functions. We demonstrate that these can be interpreted as the discrete convex conjugate of a learnable function, thus encoding the convex hull of its epigraph (the set of points lying on or above the function's graph). The contributions of this paper span theoretical proofs, technical developments, and empirical confirmations, and can be summarized in the following list:

- A novel variance-preserving initialization method is introduced for orthogonal learnable activations in neural networks. Assuming an orthonormal function basis, this method ensures that the output variances are unitary and match those of the derivative, leading to stable training.
- Empirically showing that deep neural networks like ConvNeXt (Liu et al., 2022) and GPT-2 (Radford et al., 2019) can be trained using orthogonal learnable activations for tasks like image classification on ImageNet1k (Deng et al., 2009) and language modeling on OpenWebText (Gokaslan & Cohen, 2019). The innovation eliminates the need for additional mechanisms (e.g., ReLU, SoftSign...) to maintain training stability.
- Proving in Appendix F that polynomially activated neural networks are polynomial mappings.
- Developing Hermite, Fourier, and Tropical activations, addressing finite-precision floating-point issues, and designing efficient parallel algorithms and kernels for their implementation.

## 2 RELATED WORK

The use of polynomial activations has long been denigrated, probably by the rise of works such as Pinkus (1999) and Leshno et al. (1993) which have mathematically demonstrated that the universal approximation property is equivalent to the use of a non-polynomial activation function. The Universal Approximation Theorem Cybenko (1989); Hornik et al. (1990) holds for neural networks of arbitrary width and bounded depth. However, recent work such as Kidger & Lyons (2020); Gao et al. (2025) show that in the framework of bounded width and arbitrary depth, every nonaffine continuous function is possible to use in practice, including polynomial activation functions. We show empirically in this work that polynomial activations can converge in the context of large-scale deep networks and datasets, provided coefficients are learnable, and initialization is suitable. The empirical demonstration of the effectiveness of polynomial activations made here was achieved without the use of other functions intended to regularize convergence, such as the SoftSign function borrowed from Turian et al. (2009) and used in Lokhande et al. (2020) for Hermite activations, or a ReLU function, or any normalization, as recently done in Zhuo et al. (2025). This confirmation that polynomial activations are practicable opens the way to representing deep neural networks as multivariate polynomial mappings. As in Kileel et al. (2019) and Kubjas et al. (2024), which see that these types of networks have greater expressive potential, we show in Appendix F that deep polynomially activated neural networks are indeed multivariate polynomial mappings.

The adoption of polynomial bases is further justified by recent theoretical advancements in neuroalgebraic geometry, particularly through the notions of neuromanifolds, which are the image of a polynomially activated neural network, and neurovarieties, which are defined as the closure of neuromanifolds in the Zariski topology Amari (1994); Amari et al. (2001; 2006); Calin (2020). A key advantage of polynomial activations is that they can make neural networks identifiable: under appropriate assumptions, the network parameters are uniquely determined by the represented function, up to a finite set of permutations.

Recent results by Usevich et al. (2025) provide a constructive proof for finite identifiability for MLPs with monomial activations: for a general choice of learnable parameters (i.e., outside a set of Lebesgue measure zero), the network admits a finite number of non-equivalent representations. The proof proceeds inductively, showing that if pairs of layers are identifiable, the whole network is identifiable. In particular, non-increasing-width networks are shown to be identifiable, though they may require large and linearly increasing polynomial degrees, which motivates stable training for higher-degree polynomial activations, as explored in our work. Similarly, Shahverdi et al. (2026) show that general polynomial activations in MLPs and CNNs are identifiable, and that singularities of their neuromanifolds correspond to sparse subnetworks. For MLPs, these singularities often coincide with critical points of the mean-squared error loss. Together, these results suggest that the algebraic structure of polynomials allows for a principled understanding of the loss landscape. This perspective is consistent with the findings of Ainsworth et al. (2023), who studied neural network basins and demonstrated that the weights of one trained model can be permuted to align with those of another. Such alignment effectively establishes identifiability up to permutation symmetries, making the parametrization finite-to-one.

Furthermore, quadratic forms can be interpreted as MLPs with degree-2 polynomial activations, which have recently been successfully employed in deep architectures Fan et al. (2023); Chen et al. (2025).

The subject of learnable activations has seen a resurgence thanks to the popularity enjoyed by the KAN article Liu et al. (2025). In Appendix J, we'll digress for a while to explain how these are inspired by the Kolmogorov-Arnold theorem Kolmogorov (1957). Further related work appears in Appendix K.

## 3 METHODS

### 3.1 VARIANCE PRESERVING INITIALIZATION

The variance-preserving principle He et al. (2015) mentioned in the introduction is expressed in the following. Consider an input vector $x = (x_0, \ldots, x_i, \ldots, x_{C_{in}}) \in \mathbb{R}^{C_{in}}$, $C_{in} \in \mathbb{N}^*$, where all $x_i$ are mutually independent and uniformly distributed. Preserving the variance in an MLP layer with a learnable weight tensor $W$ of inner dimension $C_{in}$ and an activation function $F$ amounts to:

$$\text{Var}[x] = C_{in} \,\text{Var}[WF(x)] \tag{1}$$

If we suppose that $x$ and $W$ are independent and of finite variance, we have:

$$\text{Var}[x] = C_{in} \left( \text{Var}[W] \cdot \mathbb{E}\left[F(x)^2\right] + \text{Var}[F(x)] \cdot \mathbb{E}\left[W\right]^2 \right) \tag{2}$$

**Assumption 3.1.** We initialize $W$ such as $\mathbb{E}[W] = 0$.

Since we always assume that $W$ is initialized with a zero mean, Eq. 2 simplifies into:

$$\text{Var}[x] = C_{in} \,\text{Var}[W] \cdot \mathbb{E}\left[F(x)^2\right] \tag{3}$$

Thus, to calculate the variance of the weights, we should calculate the following ratios:

**Definition 3.2.** The forward gain of the MLP layer is defined by:

$$\alpha = \text{Var}[x] \cdot \mathbb{E}\left[F(x)^2\right]^{-1} \tag{4}$$

Similarly, and in a backward manner,

**Definition 3.3.** The backward gain is the gain of the derivative of the activation with respect to $x$, which we denote by $F'$, and is defined as:

$$\alpha' = \text{Var}[x] \cdot \mathbb{E}\left[F'(x)^2\right]^{-1} \tag{5}$$

Since a deep neural network is essentially a composition of MLP layers, an appropriate initialization method must avoid reducing or amplifying the input signals He et al. (2015).

**Assumption 3.4.** From now on, we assume that both the input signal $x$ and its gradient $\Delta x$ follow a distribution of mean 0 and variance 1.

Therefore, calculating the gains $\alpha$ and $\alpha'$ in an MLP (or equivalently, a convolution layer) involves calculating only the inverse of the second-order moments of the activation functions and their derivatives. Interestingly, for the ReLU function, we have $\alpha = \alpha' = 2$. Hence, the scaling of the standard deviation of the weights $W$ in He et al. (2015) by a factor $\sqrt{2/C_{in}}$. More details can be found in Appendix B.

Given an arbitrary activation, equality of forward and backward gains is not always achieved by default, as in ReLU. In the next section, we show the conditions for an activation function written in an orthonormal coordinate system to verify the forward-backward gain equality. To illustrate this point, we will calculate the second moment for Hermite and Fourier basis decompositions, given their compatibility with the normal and uniform distributions, respectively.

## 3.2 VARIANCE PRESERVING INITIALIZATION FOR THE HERMITE ACTIVATION FUNCTION

**Definition 3.5.** $\forall n \in \mathbb{N}$, the probabilist Hermite polynomials can be defined as follows:

$$\mathrm{He}_n(x) = (-1)^n e^{\frac{x^2}{2}} \frac{d^n}{dx^n} e^{-\frac{x^2}{2}} \tag{6}$$

$n$ is called the degree of the Hermite polynomial, and we have the first terms:

$$\mathrm{He}_0(x) = 1 \quad \mathrm{He}_1(x) = x \quad \mathrm{He}_2(x) = x^2 - 1 \quad \mathrm{He}_3(x) = x^3 - 3x$$

Hermite polynomials constitute a suitable choice for calculating the moment of order 2 when $x$ follows a standard normal distribution $\mathcal{N}(0,1)$ as evidenced by the following property 3.6.

**Property 3.6.** $\forall m, n \in \mathbb{N}^2$, we have:

$$\int_{-\infty}^{\infty} \mathrm{He}_m(x) \, \mathrm{He}_n(x) e^{-\frac{x^2}{2}} \, dx = \sqrt{2\pi} n! \delta_{nm} \tag{7}$$

With $\delta_{nm}$, the Kronecker delta.

**Definition 3.7.** We define the Hermite activation $F \colon \mathbb{R} \to \mathbb{R}$ with its learnable coefficients $\forall n \in \mathbb{N}$, $\forall k \in [\![0, n]\!]$, $a_k \in \mathbb{R}$ as:

$$x \mapsto F(x) = \sum_{k=0}^{n} \frac{a_k}{k!} \mathrm{He}_k(x) \tag{8}$$

**Theorem 3.8.** *Variance-preserving coefficient initialization of Hermite activation. Let*

$$\forall k \in [\![1, n]\!] \; a_k = 1 \; and \; a_0 = \sqrt{1 - \frac{1}{n!}} \tag{9}$$

*Then, using this initialization, the forward and backward gains become the same and are equal to:*

$$\alpha = \alpha' = \left( \sum_{k=0}^{n-1} \frac{1}{k!} \right)^{-1} \tag{10}$$

*Proof.* The proof is provided in Appendix C. □

**Corollary 3.9.** *In the limit case $n \to +\infty$, the coefficient initialization in Theorem 3.8 could be divided by a factor $\sqrt{e}$, with $e \approx 2.7182\ldots$, in order to have unitary forward and backward gains. $\forall k \in [\![1, n]\!]$:*

$$a_k = \frac{1}{\sqrt{e}} \; and \; a_0 = \frac{1}{\sqrt{e}}\sqrt{1 - \frac{1}{n!}} \tag{11}$$

*Remark* 3.10. The choice of an orthonormal family of functions depends on the input's probability distribution. For a normally distributed input, Hermite polynomials simplify the computation of second-order moments and related gains. For a uniform distribution over $[-\pi, \pi]$, trigonometric functions (Fourier series) are appropriate. If the input follows a Wigner semi-circle distribution (of measure $\sqrt{1 - x^2} dx$), then the Chebyshev polynomials of the second kind are the suitable choice.

### 3.3 VARIANCE PRESERVING INITIALIZATION FOR THE FOURIER ACTIVATION FUNCTION

The forward and backward gains for a Hermite activation have been calculated under the assumption that the input $x$ follows a normal distribution, such that the initial coefficients provide equal gains. The subsequent analysis will establish the same result for a truncated Fourier series expansion of order $n \in \mathbb{N}$.

**Assumption 3.11.** The input $x$ is assumed now to follow a uniform distribution on the interval $[-\pi, \pi]$, with $\pi \approx 3.1415\ldots$, denoted as $x \sim \mathcal{U}(-\pi, \pi)$.

**Definition 3.12.** We consider the following Fourier activation $F: \mathbb{R} \to \mathbb{R}$:

$$x \mapsto F(x) = a_0 + \sum_{k=1}^{n} \frac{(a_k \cos(kx) + b_k \sin(kx))}{k!} \tag{12}$$

where $(a_k)_{k \in \mathbb{N}}$ and $(b_k)_{k \in \mathbb{N}^*}$ are real learnable coefficients.

**Theorem 3.13.** *Variance-preserving coefficient initialization of Fourier activation. Let*

$$\forall k \in [\![1, n]\!] \ a_k = 1, \ b_k = 1, \ and \ a_0 = \sqrt{1 - \frac{1}{(n!)^2}} \tag{13}$$

*Then, using this initialization, the forward and backward gains become the same and are equal to:*

$$\alpha = \alpha' = \left( \sum_{k=0}^{n-1} \frac{1}{(k!)^2} \right)^{-1} \tag{14}$$

*Proof.* The proof is provided in Appendix D. $\square$

**Corollary 3.14.** *In the limit case $n \to +\infty$, in order to have unitary forward and backward gains, the coefficient initialization in Theorem 3.13 could be divided by a factor $\sqrt{I_0(2)}$, with $I_\alpha(x)$ is the modified Bessel function of the first kind of order $\alpha$, and we have $I_0(2) \approx 2.2795\ldots \forall k \in [\![1, n]\!]$ :*

$$a_k = \frac{1}{\sqrt{I_0(2)}}, \ b_k = \frac{1}{\sqrt{I_0(2)}}, and \ a_0 = \frac{1}{\sqrt{I_0(2)}} \sqrt{1 - \frac{1}{(n!)^2}} \tag{15}$$

*Remark* 3.15. In both Definitions 3.7 and 3.12, the terms inside the sum are scaled by a factor of $k!$, yielding exponential series. In practice, it is possible to scale the terms using other converging series such as $k^p$ with $p > 1$. We experimented with this last alternative and observed no statistically significant impact on loss convergence, though we did observe better stability for higher polynomial degrees in the exponential variant.

### 3.4 VARIANCE PRESERVING INITIALIZATION FOR THE TROPICAL ACTIVATION FUNCTION

**Definition 3.16.** The max-tropical semiring $\mathbb{T}$ is the semiring $\mathbb{T} = (\mathbb{R} \cup \{+\infty\}, \oplus, \otimes)$, with the operations, $\forall x, y \in \mathbb{R} \cup \{+\infty\}^2$:

$$x \oplus y := \max\{x, y\} \quad \text{and} \quad x \otimes y := x + y \tag{16}$$

Equivalently, we could define the min-tropical semiring by substituting the max operation in $\oplus$ with a min operation. By extension, we define for all $a \in \mathbb{N}$ the tropical power of $x$ raised to $a$ as multiplying $x$ to itself $a$ times:

$$x^{\otimes a} := x \otimes \cdots \otimes x = a \cdot x \tag{17}$$

**Definition 3.17.** The *tropicalization* of a polynomial of degree $n \in \mathbb{N}$ is defined as $F: \mathbb{R} \mapsto \mathbb{R}$, with $\forall n \in \mathbb{N}, \forall k \in [\![0, n]\!] \ a_k \in \mathbb{R}$ are the polynomial learnable coefficients:

$$x \mapsto F(x) = \bigoplus_{k=0}^{n} a_k \otimes x^{\otimes k} := \max_{k=0}^{n} \{a_k + kx\} \tag{18}$$

With $\max_{k=0}^{n} \{a_k + kx\} := \max(a_0, a_1 + x, \ldots, a_n + nx)$.

**Definition 3.18.** *Convex conjugate (Legendre-Fenchel).* Let $x \in \mathbb{R}$, $f^* \colon \mathbb{R} \to \mathbb{R}$ is the convex conjugate of $f \colon \mathbb{R} \to \mathbb{R}$ if and only if:

$$f^*(x) = \sup_{k \in \mathbb{R}} \{kx - f(k)\} \tag{19}$$

**Theorem 3.19.** *Variance-preserving coefficient initialization of Tropical activation. Let*

$$\forall k \in [\![0, n]\!] \, a_k = 1 \tag{20}$$

*Then, applying this initialization to the limit case of $n \to \infty$ yields an equal unitary gain both forward and backward for the following "scaled" definition of the tropical activation:*

$$x \mapsto F(x) = \frac{\sqrt{2}}{n} \max_{k=0}^{n} \{a_k + kx\} \tag{21}$$

*Proof.* The proof is provided in Appendix E . □

The tropical polynomial activation can be viewed as a generalization of the ReLU activation. Furthermore, it can be interpreted as a discrete version of the convex conjugate of a function $f$ whose values at the natural integers $k \in \mathbb{N}$ are $f(k) = -a_k$ effectively encoding the convex hull of the epigraph of $f$, as illustrated in Figures 5 and 6.

## 3.5 PRACTICAL IMPLEMENTATION

In what follows, we outline the considerations we have taken to implement Hermite, Fourier, and Tropical polynomial activations efficiently in PyTorch.

**Weight decay.** An important aspect of training learnable activations is that their learnable coefficients should be trained without weight decay, as it could bias them toward zero.

**Explicit Hermite formula.** We can show by induction that the following definition is equivalent to the one in Eq. 6:

$$\frac{\mathrm{He}_n(x)}{n!} = \sum_{m=0}^{\lfloor \frac{n}{2} \rfloor} \frac{(-1)^m}{m!(n-2m)!} \frac{x^{n-2m}}{2^m} \tag{22}$$

We can see that the formula 22 can be parallelized, and is, therefore, the core of the algorithm we have developed in native PyTorch to compute Hermite activations (see Algorithm 2).

**A dedicated Hermite kernel.** Along with the parallel implementation of the Hermite activation, we developed a dedicated kernel that leverages the derivation established in C.5 for the backward pass exploiting the fact that the derivative of a polynomial is a polynomial of lower degree and the following recurrence formula in the forward pass to optimize performance and memory usage (see Algorithm 3):

$$\mathrm{He}_{n+1}(x) = x \, \mathrm{He}_n(x) - n \, \mathrm{He}_{n-1}(x) \tag{23}$$

**Alternative Fourier formula.** The definition of Fourier activation given in Definition 3.12 is under the Sine-Cosine form. In practice, we use the following equivalent Amplitude-Phase formulation (see Algorithm 5):

$$x \mapsto F(x) = a_0 + \sqrt{2} \sum_{k=1}^{n} \frac{a_k \cos(f_k x - \phi_k)}{k!} \tag{24}$$

as it is less expensive in terms of FLOP. The learnable parameters here are initialized as follows: $\forall k \in \mathbb{N}^* f_k = k, \phi_k = \frac{\pi}{4}$ and $a_k$ and $a_0$ initialized as in 3.14. In our implementation of Fourier activation, not only were the coefficients learnable, but also the frequencies, yielding to what is known as a "cosine basis" Mallat (2009) rather than the Fourier series.

**Initializing by fitting a classical activation Function.** Using a family of orthonormal functions permits an easy calculation of the initialization gain without resorting to the trick of fitting a function to an activation whose gain is known or easy to calculate, as in Yang & Wang (2025) with Safe Padé activation Molina et al. (2020). However, in some cases, such as continuing or fine-tuning a model that was pretrained with a classical activation, using one of the learnable activations presented here to fit a classical activation could still be relevant. By *fitting* we mean performing a Lagrange interpolation.

This could be accomplished via a direct method involving the inversion of a Vandermonde matrix (Lagrange, or Newton's methods), or by an iterated gradient descent method (Gauss-Jordan method).

Two precautions need to be taken, however, when performing such interpolation. The first concerns the maximum degree that should be considered in order to fit the function on a given interval. Figure 1 (left) shows how far a Hermite activation of degree 3 can be accurately fitted, while Figure 1 (right) shows the extent to which a Hermite activation of degree 8 can be accurately fitted. The second precaution concerns the derivative of the activation with respect to the derivative of the target function to be interpolated. A Lagrange interpolation of a function is not always sufficient to fit its $k$-th derivatives. If we want to interpolate a function and its derivative(s) simultaneously, we refer to this as a Hermite interpolation.

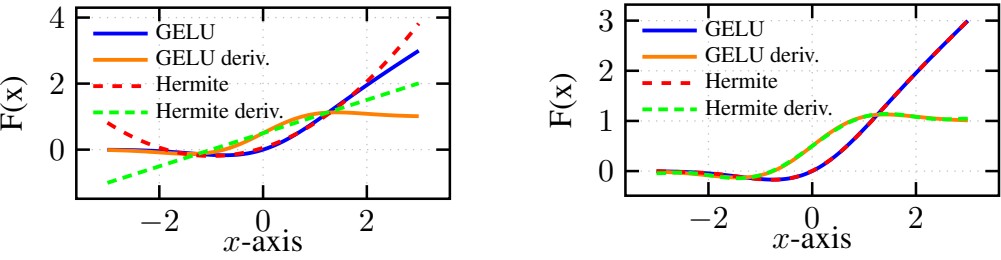

Figure 1: Fitting a GELU with a Hermite Activation of degree 3 (left) and of degree 8 (right).

In the case of the Fourier activation, we observe in Figure 2 (left) that a Lagrange interpolation is not sufficient and that higher-order frequencies occur in the derivative approximation. This phenomenon can be likened to aliasing and can be circumvented by performing a simple Hermite interpolation instead of a Lagrange interpolation, as shown in Figure 2 (right). Berrut & Welscher (2007) examined the exact solutions to this last problem for trigonometric functions.

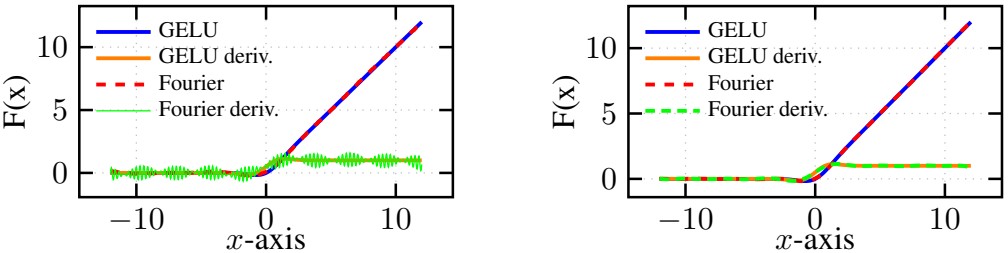

Figure 2: Lagrange interpolation (left) and Hermite interpolation (right) of a GELU with a Fourier Activation of degree 6.

The success in fitting classical activations with Padé approximants in Yang & Wang (2025) could be attributed to the fact that a Padé approximant is by definition the rational function that coincides with a function to be interpolated to the highest possible order, thus naturally achieving a Hermite interpolation. A good fit of a non-convex function by a tropical polynomial activation is impossible since tropical polynomials are convex by definition. Therefore, in Appendix H we show how rational tropical activations (an extension of tropical polynomials) could, in principle, achieve this fitting.

## 4 EXPERIMENTS

### 4.1 PRELIMINARY IMAGE CLASSIFICATION RESULTS ON CIFAR10

We trained ConvNeXt-T (Liu et al., 2022) on CIFAR-10 (Krizhevsky et al., 2009) for 300 epochs, averaging results over 10 random seeds. The experimental setup and results for CIFAR10 classification can be found in Appendix L. The three proposed learnable activations consistently outperformed baseline activations on test metrics. Results are shown in Table 7 and Figures 7, 8, and 9.

## 4.2 DECISION BOUNDARIES ON NOISY CLASSIFICATION DATASETS

We compared the decision boundaries of four single-layer neural networks trained on a simple, noisy classification dataset, each using a different activation function to evaluate how the choice of activation function affects classification behavior and boundary smoothness. Details of the visualizations of decision boundaries on multiple noisy datasets are provided in Appendix M.

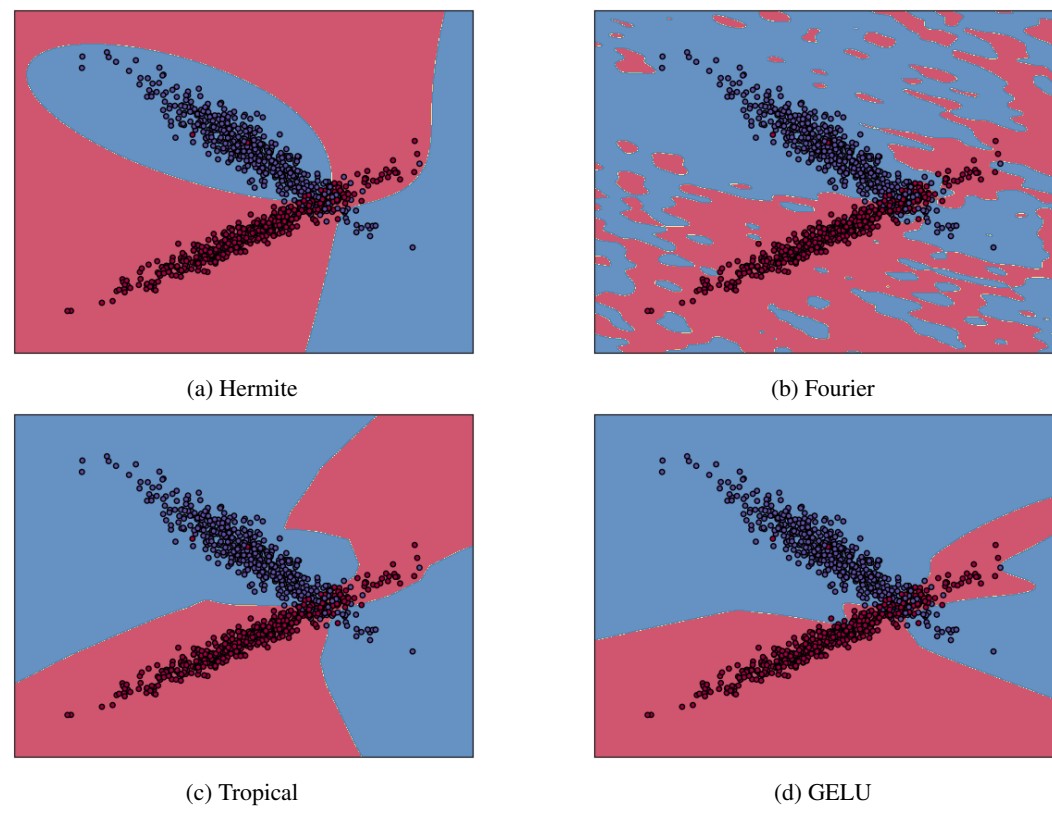

(a) Hermite

(b) Fourier

(c) Tropical

(d) GELU

Figure 3: Decision boundaries for different activation functions

## 4.3 VISION TASK: CONVNEXT-T IMAGE CLASSIFICATION ON IMAGENET1K

We evaluated the ConvNeXt-T model on the ImageNet1k dataset Deng et al. (2009) for single-class image classification. The baseline ConvNeXt-T model employed GELU as the activation function in its MLP blocks. To analyze the impact of our learnable activations, we replaced GELU with Hermite polynomial, Fourier trigonometric, and Tropical polynomial activation functions under our proposed initialization scheme. Each model was trained under identical conditions with fixed random seeds to ensure reproducibility and comparability. The evaluation metrics included: training loss, Top-1 and Top-5 validation accuracy. Table 1 and Figures 11, 12, and 13 summarize our results. We reproduced all experiments using five different random seeds. For each trial, we report the mean ± standard deviation at a fixed epoch. The experimental setup followed the approach and hyperparameter configuration detailed in Liu et al. (2022).

**Ablation Studies.** Additionally, ablation studies were performed on this vision task to establish the impact of the degree for the learnable activations (Table 3), the impact of our proposed initialization scheme (Table 4), and if making the activation coefficients learnable was useful (Table 5). Higher degrees generally improved performance, with all proposed activations showing consistent improvements in Top-1 and Top-5 accuracy as the degree increased. Furthermore, making activation coefficients learnable consistently resulted in better performance across all activation functions. Initialization with the proposed method led to improvements, especially for Hermite activation, where our derived initialization scheme outperformed GELU-based initialization.

Table 1: Training and validation results of ConvNeXt-T (28M) model on ImageNet-1k classification. Values are reported as mean ± standard deviation over 5 seeds. p-values (two-tailed Student's t-test assuming equal variances) are for Val Top-1 accuracy compared to GELU.

| Act. | Deg. | Train Loss ↓ | Val Top-1(%) ↑ | Val Top-5(%) ↑ | FLOP | FLOP/Act. | p-value (Top-1) |
|------|------|--------------|----------------|----------------|------|-----------|-----------------|
| GELU | - | 2.824 ± 0.0051 | 82.06 ± 0.072 | 95.92 ± 0.038 | 4.57G | 12 | - |
| Tropical | 6 | 2.854 ± 0.0080 | 82.17 ± 0.063 | 95.95 ± 0.072 | 4.62G | 3d + 1 = 19 | 0.0345 (*) |
| Fourier | 6 | **2.759 ± 0.0167** | 81.64 ± 0.153 | 95.47 ± 0.049 | 4.83G | 7d + 1 = 43 | 0.0005 (***) |
| Hermite | 3 | 2.788 ± 0.0072 | **82.22 ± 0.064** | **95.97 ± 0.045** | 4.58G | 4d + 1 = 13 | 0.0062 (**) |

## 4.4 LANGUAGE TASK: GPT-2 (124M) NEXT TOKEN PREDICTION ON OPENWEBTEXT

For the language modeling task, we trained the GPT-2 model Radford et al. (2019) on the OpenWeb-Text dataset Gokaslan & Cohen (2019) for next-token prediction. The baseline GPT-2 used GELU activation, and we compared it against SiLU (Elfwing et al., 2018), Hermite, Fourier, and Tropical activations under our proposed initialization scheme. All models were trained with identical hyperparameters and initialization seeds to ensure consistent and reproducible comparisons. The evaluation metrics included: training and test losses and perplexities (which are simply the exponential of the loss). Table 2 and Figures 14 and 15 summarize our results. We reproduced all experiments using five different seeds. For each trial, we report the mean ± standard deviation at a fixed iteration. The experimental design followed the guidelines established in Radford et al. (2019) and the open source reproduction available at Karpathy (2022). We used a total batch size of $786,432$ of which a context length of $1024$ tokens for a total of $210,000$ iterations.

Table 2: Training and validation results for next-token prediction using GPT-2 (124M) model with different activations. Values are reported as mean ± standard deviation over 5 different seeds. Perplexity is computed as $\exp(\text{loss})$. p-values (two-tailed Student's t-test assuming equal variances) compare each activation's validation loss against GELU.

| Act. | Deg. | Train PPL ↓ | Train Loss ↓ | Val PPL ↓ | Val Loss ↓ | FLOP | p-value (Val Loss) |
|------|------|-------------|--------------|-----------|------------|------|--------------------|
| GELU | - | 19.003 ± 0.156 | 2.944 ± 0.0082 | 19.319 ± 0.076 | 2.961 ± 0.0039 | 87.52G | - |
| SiLU | - | 19.324 ± 0.106 | 2.962 ± 0.0055 | 19.664 ± 0.088 | 2.979 ± 0.0045 | 87.37G | 0.0001 (***) |
| Tropical | 6 | 18.840 ± 0.107 | 2.936 ± 0.0057 | 19.027 ± 0.055 | 2.946 ± 0.0029 | 87.75G | 0.0001 (***) |
| Fourier | 6 | 18.761 ± 0.071 | 2.930 ± 0.0038 | 18.965 ± 0.154 | 2.941 ± 0.0086 | 88.69G | 0.0014 (**) |
| Hermite | 3 | **18.678 ± 0.093** | **2.926 ± 0.0049** | **18.821 ± 0.293** | **2.932 ± 0.0175** | 87.56G | 0.0067 (**) |

All experiments were conducted under fixed configurations to ensure that any observed differences were solely due to the choice of activation function, allowing for fair and reproducible comparisons[2].

## 4.5 FINETUNING EXPERIMENT ON CIFAR10

Using the insights from Sec. 3.5, we conducted a fine-tuning experiment in a transfer learning setting. Specifically, we investigated whether initializing a learnable activation by fitting a classical one, using Hermite interpolation, can improve performance when adapting a pretrained model to a new dataset. This experiment complements our theoretical analysis by demonstrating how fitting classical activations can serve as an effective initialization strategy. The experimental procedure and results for activation finetuning are available in Appendix O.

## 5 PARAMETERS, MEMORY, FLOP COUNT, AND EXECUTION TIME

The proposed activation functions introduce a negligible number of additional parameters. For example, Hermite activations of degree $d = 3$ add only 72 parameters to ConvNeXt-Tiny (28M total), corresponding to 0.0002%, with similarly minimal overheads for Tropical and Fourier activations. Hermite activations leverage a recursive formulation (Alg. 3) that reduces both FLOP and required

---

[2]The code to reproduce the experiments is available at: https://github.com/K-H-Ismail/torchortho

memory (vRAM) complexity from $\mathcal{O}(d^2)$ to $\mathcal{O}(d)$, requiring only simple arithmetic per term. Fourier and Tropical activations also scale linearly with degree ($\mathcal{O}(d)$), as illustrated in Figure 17 and Table 9, measured on CPU. On GPUs, smaller degrees benefit from vectorized computation, leading to reduced runtime and near-constant $\mathcal{O}(1)$ scaling for low degrees (Figure 18, Table 10).

We further evaluated average training times per epoch across varying MLP widths and depths (Table 11). The proposed activations can incur higher latency compared to GELU in deep networks, but are often faster in shallower ones. Slowdowns relative to GELU were analyzed across widths (Figure 19) and depths (Figure 20). Slowdowns are largely independent of width but increase approximately linearly with depth, with Hermite activations showing the largest slope, followed by Fourier and Tropical. This suggests that the proposed activations are more suitable for shallow, wide MLPs. This observation aligns with Appendix F, where a polynomially activated MLP of arbitrary depth is shown to be equivalent to a high-degree single-layer multivariate polynomial.

# 6 DISCUSSION

The results presented in this paper demonstrate the potential of using learnable activation functions based on orthogonal function bases and tropical polynomials in large-scale neural network tasks. Our experiments on ImageNet-1K and OpenWebText with deep models such as ConvNeXt and GPT-2 show for the first time that such activations can lead to improvements over traditional static functions like ReLU and GELU, both in terms of image classification and language modeling.

This challenges the long-standing notion that polynomial activations are inherently unsuitable for deep learning, as demonstrated by prior work. Our approach provides empirical evidence that, with appropriate initialization, polynomial activations can indeed be competitive. One of the key takeaways from our findings is the effectiveness of our proposed variance-preserving initialization scheme. The choice of orthogonal functions plays an essential role in achieving a closed-form expression for the second-order moment. Furthermore, the use of tropical polynomials, which are not orthogonal, introduces a FLOP-light alternative approach to polynomial activations.

While our approach shows promise, there are several avenues for future exploration. Extending the framework to other activation families, such as wavelets, is straightforward. Multiplying the Hermite activation presented in this work by the term $\exp\left(-x^2/2\right)$ gives what is known as Hermitian wavelets Brackx et al. (2008), and applying the same to the Fourier activation yields the Morlet wavelet Grossmann & Morlet (1984) (or Gabor wavelet Gabor (1946)). Wavelets retain good orthogonal properties with respect to the adequate scalar product, and the calculation of the second moment is slightly modified to take account of the additional decaying exponential term. Using wavelet activations instead of polynomials could enhance variance stability by providing finite function support, with potential bio-plausibility implications. By expressing a Fourier series in its complex form, a network with Fourier activation can be viewed as a complex-valued neural network, offering a framework for modeling neuronal synchronization through the phase and amplitude relationships of oscillatory brain activity. Extension to other non-orthogonal functions, such as rational functions, could be done, for example, by means of a Laplace transform of the Fourier activation.

# 7 CONCLUSION

In this work, we introduced a novel framework for integrating learnable activation functions based on orthogonal function bases and tropical polynomials into deep neural networks, addressing challenges like variance preservation and stable gradient flow. Extensive experiments with the ConvNeXt model on ImageNet1k and the GPT-2 model on OpenWebText showed that learnable polynomial activations match or exceed traditional activation functions during large-scale training and fine-tuning on smaller tasks, demonstrating their practical viability and challenging conventional beliefs about polynomial activations in neural networks. Importantly, we adopt a broad notion of "polynomial", encompassing not only classical algebraic polynomials but also trigonometric expansions and tropical polynomials. Our results pave the way for representing deep neural networks as polynomial mappings whose hypothesis classes correspond to geometric objects akin to algebraic varieties, or, in the tropical setting, piecewise-linear polyhedral complexes, with future work focused on exploring a careful relaxation of these last.

ACKNOWLEDGEMENTS

In alphabetical order, we thank Emile de Bruyn, Jan Ebert, Jiangtao Wang, and Oleg Filatov for their helpful discussions and feedback on this manuscript. We also thank David Guo and @iiisak for helpful technical discussions. This work would not have been possible without financial and computational support. This research was supported by the German Federal Ministry for Economic Affairs and Climate Action through the project "NXT GEN AI METHODS" and by the Helmholtz AI Cooperation Unit (HAICU) through the Helmholtz Foundation Model Initiative as a part of the Synergy Unit. We gratefully acknowledge the Gauss Centre for Supercomputing e.V. (www.gauss-centre.eu) for funding this project by providing computing time on the Supercomputers JUWELS and JURECA at Jülich Supercomputing Centre (JSC).

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

## A    SCHEMATIC OF BASIS-MLP

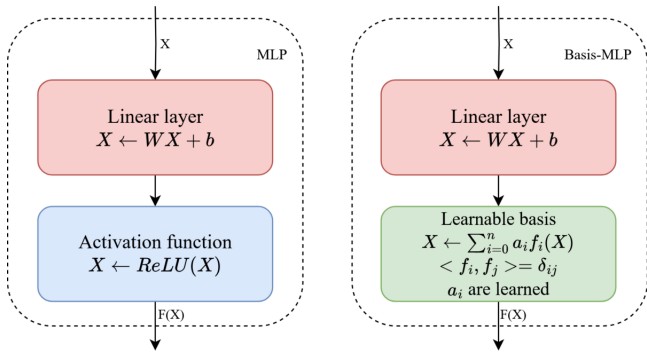

Figure 4: A classical MLP (linear + ReLU) vs Basis-MLP (linear + learnable basis function) blocks.

## B    FORWARD AND BACKWARD SECOND MOMENT CALCULATION FOR THE RELU ACTIVATION FUNCTION

### B.1    SECOND MOMENT OF THE RELU ACTIVATION FUNCTION

The Rectified Linear Unit (ReLU) activation function Nair & Hinton (2010), defined as:

$$\text{ReLU}(x) = \max(0, x) \tag{25}$$

is commonly used in neural networks due to its simplicity and effective gradient propagation. When $x$ is drawn from a standard normal distribution $x \sim \mathcal{N}(0, 1)$, the second moment of the ReLU function is:

$$\mathbb{E}[\text{ReLU}(x)^2] = \int_0^\infty x^2 \frac{1}{\sqrt{2\pi}} e^{-x^2/2} dx = \frac{1}{2} \tag{26}$$

### B.2    SECOND MOMENT OF THE DERIVATIVE OF RELU

The derivative of ReLU, given by:

$$\frac{d}{dx}\text{ReLU}(x) = \begin{cases} 1, & x > 0, \\ 0, & x \leq 0, \end{cases} \tag{27}$$

acts as a binary indicator of positive inputs. The second moment of this derivative when $x \sim \mathcal{N}(0, 1)$ is:

$$\mathbb{E}\left[\left(\frac{d}{dx}\text{ReLU}(x)\right)^2\right] = \int_0^\infty \frac{1}{\sqrt{2\pi}} e^{-x^2/2} dx = \frac{1}{2} \tag{28}$$

This result matches the variance of the ReLU function itself and validates the gain of 2 for variance-preserving weight initialization with ReLU activations.

## C    PROOF OF THE THEOREM 3.8

**Definition C.1.** We define the Hermite activation $F \colon \mathbb{R} \to \mathbb{R}$ with its learnable coefficients $\forall n \in \mathbb{N}$, $\forall k \in [\![0, n]\!]$ $a_k \in \mathbb{R}$ as:

$$x \mapsto F(x) = \sum_{k=0}^n \frac{a_k}{k!} \text{He}_k(x) \tag{29}$$

**Property C.2.** $\forall m, n \in \mathbb{N}^2$, *we have:*

$$\int_{-\infty}^\infty \text{He}_m(x) \text{He}_n(x) e^{-\frac{x^2}{2}} dx = \sqrt{2\pi} n! \delta_{nm} \tag{30}$$

*With $\delta_{nm}$ the Kronecker delta.*

**Proposition C.3.** *The second moment of this activation with respect to $\mathcal{N}(0, 1)$ is:*

$$\mathbb{E}\left[F(x)^2\right] = \sum_{k=0}^{n} \frac{a_k^2}{k!} \tag{31}$$

*Proof.* The proof relies on the orthonormality property C.2.

The orthonormality property C.2 means that: $\forall m, n \in \mathbb{N}^2$,

$$\int_{-\infty}^{\infty} \frac{\mathrm{He}_n(x)^2}{n!} \frac{e^{-\frac{x^2}{2}}}{\sqrt{2\pi}} dx = 1 \tag{32}$$

and if $m \neq n$

$$\int_{-\infty}^{\infty} \mathrm{He}_m(x) \, \mathrm{He}_n(x) \frac{e^{-\frac{x^2}{2}}}{\sqrt{2\pi}} dx = 0 \tag{33}$$

Given the definition (Def. C.1) of a Hermite activation $F$, we have:

$$\mathbb{E}\left[F(x)^2\right] = \int_{-\infty}^{+\infty} F^2(x) \frac{e^{-\frac{x^2}{2}}}{\sqrt{2\pi}} dx \tag{34}$$

$$= \int_{-\infty}^{+\infty} \left(\sum_{k=0}^{n} \frac{a_k}{k!} \mathrm{He}_k(x)\right)^2 \frac{e^{-\frac{x^2}{2}}}{\sqrt{2\pi}} dx \tag{35}$$

Using the orthogonal property Eq. 33, the cross terms cancel out, and we have:

$$\mathbb{E}\left[F(x)^2\right] = \int_{-\infty}^{+\infty} \sum_{k=0}^{n} \frac{a_k^2}{(k!)^2} \mathrm{He}_k(x)^2 \frac{e^{-\frac{x^2}{2}}}{\sqrt{2\pi}} dx \tag{36}$$

$$= \sum_{k=0}^{n} \frac{a_k^2}{(k!)^2} \int_{-\infty}^{+\infty} \mathrm{He}_k(x)^2 \frac{e^{-\frac{x^2}{2}}}{\sqrt{2\pi}} dx \tag{37}$$

Given the normality property Eq. 32, we have

$$\mathbb{E}\left[F(x)^2\right] = \sum_{k=0}^{n} \frac{a_k^2}{k!} \int_{-\infty}^{+\infty} \frac{\mathrm{He}_k(x)^2}{k!} \frac{e^{-\frac{x^2}{2}}}{\sqrt{2\pi}} dx \tag{38}$$

$$= \sum_{k=0}^{n} \frac{a_k^2}{k!} \tag{39}$$

$\square$

Having found the initialization gain for the activation $F$, we now need to enforce this same gain for its derivative. Indeed, we are going to use the gradient descent algorithm to train our learnable activation networks, and having an activation gradient of high (respectively low) variance could lead to exploding (respectively vanishing) gradients, a nondesirable property for deep neural networks trained with gradient backpropagation.

**Property C.4.** *The following recurrence property is derived directly from the equation 6.* $\forall k \in \mathbb{N}$ $\forall x \in \mathbb{R}$:

$$\mathrm{He}'_k(x) = x \, \mathrm{He}_k(x) - \mathrm{He}_{k+1}(x) \tag{40}$$

**Property C.5.** *The following property is shown by induction and by using the previous property C.4.* $\forall k \in \mathbb{N}^* \forall x \in \mathbb{R}$:

$$\mathrm{He}'_k(x) = k \, \mathrm{He}_{k-1}(x) \tag{41}$$

**Proposition C.6.** *Using the last property and by the linearity of the integral, the derivative of $F$ (Def. C.1), $F' : \mathbb{R} \to \mathbb{R}$ is written as follows:*

$$x \mapsto F'(x) = \sum_{k=1}^{n} \frac{a_k}{(k-1)!} \mathrm{He}_{k-1}(x) \tag{42}$$

*Remark* C.7. A first remark here is that $\forall n > 2$: $F'$ is unbounded ($\lim_{x \to \infty} F'(x) \to \infty$). This means that $F$ is not Lipschitz continuous. Lipschitz continuity is often desired (or even required) when training a deep neural network using gradient backpropagation. However, by a suitable initial choice of the coefficients $(a_k)_{k \in [\![0,n]\!]}$ we can keep the Lipschitz constant under control.

**Proposition C.8.** *The second moment of the derivative of the Hermite activation is:*

$$\mathbb{E}\left[F'(x)^2\right] = \sum_{k=1}^{n} \frac{a_k^2}{(k-1)!} \tag{43}$$

*Proof.* Knowing that $\forall k \in \mathbb{N}^* \; \forall x \in \mathbb{R}$:

$$\mathrm{He}_k'(x) = k\,\mathrm{He}_{k-1}(x) \tag{44}$$

The definition of $F'$ becomes:

$$F' : \mathbb{R} \to \mathbb{R}$$

$$x \mapsto F'(x) = \sum_{k=1}^{n} \frac{ka_k}{k!}\,\mathrm{He}_{k-1}(x) \tag{45}$$

Thus, the second-order moment of $F'$ is:

$$\mathbb{E}\left[F'(x)^2\right] = \int_{-\infty}^{+\infty} \left(\sum_{k=1}^{n} \frac{a_k}{(k-1)!}\,\mathrm{He}_{k-1}(x)\right)^2 \frac{e^{-\frac{x^2}{2}}}{\sqrt{2\pi}}\,dx \tag{46}$$

By the orthogonal property Eq. 33, the cross terms cancel out, and we have:

$$\mathbb{E}\left[F'(x)^2\right] = \int_{-\infty}^{+\infty} \sum_{k=1}^{n} \frac{a_k^2}{((k-1)!)^2}\,\mathrm{He}_{k-1}(x)^2 \frac{e^{-\frac{x^2}{2}}}{\sqrt{2\pi}}\,dx \tag{47}$$

$$= \sum_{k=1}^{n} \frac{a_k^2}{(k-1)!} \int_{-\infty}^{+\infty} \frac{\mathrm{He}_{k-1}(x)^2}{(k-1)!} \frac{e^{-\frac{x^2}{2}}}{\sqrt{2\pi}}\,dx \tag{48}$$

$$\tag{49}$$

By the normality property Eq. 32, we finally have:

$$\mathbb{E}\left[F'(x)^2\right] = \sum_{k=1}^{n} \frac{a_k^2}{(k-1)!} \tag{50}$$

$\square$

**Proposition C.9.** *Equality between propositions C.3 and C.8 imposes that:*

$$a_0^2 = \sum_{k=1}^{n} \frac{(k-1)}{k!} a_k^2 \tag{51}$$

$$= \sum_{k=1}^{n} \left(\frac{1}{(k-1)!} - \frac{1}{k!}\right) a_k^2 \tag{52}$$

To satisfy the forward-backward gain equality, we could initialize the coefficients $(a_k)_{k \in [\![0,n]\!]}$ such as $\forall n \in \mathbb{N}^*$:

$$\forall k \in [\![1, n]\!] \; a_k = 1 \quad \text{and} \quad a_0 = \sqrt{1 - \frac{1}{n!}} \tag{53}$$

This initialization works in practice for all $n$. Furthermore, as the term $\frac{1}{n!}$ in $a_0$ vanishes quickly with $n \to +\infty$, for larger $n$ we could initialize all the coefficients to 1 including $a_0$.

In the limit case, by a simple injection of $a_k = 1$ in Prop. C.9 and then in Prop. C.8, we obtain the result.

# D   PROOF OF THE THEOREM 3.13

**Definition D.1.** We consider the following Fourier activation $F \colon \mathbb{R} \to \mathbb{R}$:

$$x \mapsto F(x) = a_0 + \sum_{k=1}^{n} \frac{(a_k \cos(kx) + b_k \sin(kx))}{k!} \tag{54}$$

where $(a_k)_{k \in \mathbb{N}}$ and $(b_k)_{k \in \mathbb{N}^*}$ are real learnable coefficients.

**Property D.2.** *The equivalent of the C.2 property for trignometric functions is given by $\forall m, n \in \mathbb{Z}^2$:*

$$\begin{cases} \int_{-\pi}^{\pi} \cos(mx) \cos(nx) dx = \pi \delta_{nm} \\ \int_{-\pi}^{\pi} \sin(mx) \sin(nx) dx = \pi \delta_{nm} \\ \int_{-\pi}^{\pi} \cos(mx) \sin(nx) dx = 0 \end{cases} \tag{55}$$

*With $\delta_{nm}$ the Kronecker delta function.*

**Proposition D.3.** *The second moment of this activation is:*

$$\mathbb{E}[F(x)^2] = a_0^2 + \frac{1}{2} \sum_{k=1}^{n} \frac{(a_k^2 + b_k^2)}{(k!)^2} \tag{56}$$

*Proof.* The proof relies on the orthonormality property D.2.

The random variable $x$ is assumed to follow a uniform distribution on the interval $[-\pi, \pi]$, denoted as:

$$x \sim \mathcal{U}(-\pi, \pi) \tag{57}$$

To compute the second moment of the Fourier activation $F(x)$, we need to compute the expected value of $F(x)^2$:

$$\mathbb{E}[F(x)^2] = \int_{-\pi}^{\pi} F(x)^2 p(x) \, dx \tag{58}$$

where $p(x)$ is the probability density function (PDF) of the uniform distribution:

$$p(x) = \frac{1}{2\pi}, \quad x \in [-\pi, \pi] \tag{59}$$

Taking the square of the definition in Eq. 54 gives:

$$F(x)^2 = \left( a_0 + \sum_{k=1}^{n} \frac{(a_k \cos(kx) + b_k \sin(kx))}{k!} \right)^2 \tag{60}$$

Using the orthogonal property D.2 and the linearity of the integral, we have:

$$\mathbb{E}[F(x)^2] = a_0^2 + \frac{1}{2\pi} \sum_{k=1}^{n} \frac{1}{(k!)^2} \int_{-\pi}^{\pi} a_k^2 \cos^2(kx) + b_k^2 \sin^2(kx) \, dx \tag{61}$$

$$= a_0^2 + \frac{1}{2\pi} \sum_{k=1}^{n} \frac{a_k^2}{(k!)^2} \left( \frac{\sin(2\pi k)}{2k} + \pi \right) + \frac{b_k^2}{(k!)^2} \left( \pi - \frac{\sin(2\pi k)}{2k} \right) \tag{62}$$

The second moment simplifies to:

$$\mathbb{E}[F(x)^2] = a_0^2 + \frac{1}{2} \sum_{k=1}^{n} \frac{(a_k^2 + b_k^2)}{(k!)^2} \tag{63}$$

$\square$

Next, we compute the second moment of the derivative of the Fourier activation $F'$. The derivative of $F$ is given by:

**Proposition D.4.** *The derivative of the Fourier activation $F' \colon \mathbb{R} \to \mathbb{R}$ is given by:*

$$F'(x) = \sum_{k=1}^{n} \frac{1}{(k-1)!} \left( -a_k \sin(kx) + b_k \cos(kx) \right) \tag{64}$$

*Remark* D.5. Contrary to the remark in C.7, $F'$ is bounded.

$$\forall x \in \mathbb{R} \colon |F'(x)| \leq \max(|a_k|, |b_k|)_{k \in [\![1,n]\!]} \sum_{k=1}^{n} \frac{1}{(k-1)!} \leq e \max(|a_k|, |b_k|)_{k \in [\![1,n]\!]} \tag{65}$$

This means that in the case of a Fourier activation, $F$ is Lipschitz continuous.

**Proposition D.6.** *The second moment of the derivative of the Fourier activation is:*

$$\mathbb{E}\left[ F'(x)^2 \right] = \frac{1}{2} \sum_{k=1}^{n} \frac{1}{((k-1)!)^2} (a_k^2 + b_k^2) \tag{66}$$

*Proof.* An orthonormality argument as for the proof in the forward case suffices. $\qquad \square$

**Proposition D.7.** *Equality between D.3 and D.6 imposes that:*

$$a_0^2 = \frac{1}{2} \sum_{k=1}^{n} \frac{(k^2-1)}{(k!)^2} (a_k^2 + b_k^2) \tag{67}$$

To satisfy the forward-backward gain equality, we could again initialize the coefficients such as $\forall n \in \mathbb{N}^*$:

$$\forall k \in [\![1,n]\!] \; a_k = b_k = 1 \text{ and } a_0 = \sqrt{1 - \frac{1}{(n!)^2}} \tag{68}$$

This initialization works in practice for all $n$. Furthermore, as the term $\frac{1}{(n!)^2}$ in $a_0$ vanishes quickly with $n \to +\infty$, for larger $n$ we could initialize all the coefficients to 1 including $a_0$.

In the limit case, by a simple injection of $a_k = 1$ in Prop. D.7 and then in Prop. D.6, we obtain the result.

*Remark* D.8. For an input $x$ of distribution $x \sim \mathcal{U}(-\sqrt{3}, \sqrt{3})$, which has a variance of $\mathrm{Var}[x] = 1$ and which is more in line with deep neural networks that seek a unitary variance preserving property across layers, we could rescale the fundamental frequency given in the definition of $F$ in Def. D.1 by redefining it as:

$$x \mapsto F(x) = a_0 + \sum_{k=1}^{n} \frac{1}{k!} \left( a_k \cos(k \frac{\pi}{\sqrt{3}} x) + b_k \sin(k \frac{\pi}{\sqrt{3}} x) \right) \tag{69}$$

The computation of the second moment stays the same except for a factor $\frac{\pi}{\sqrt{3}}$. In general if $x \sim \mathcal{U}(-l, l)$, $l \in \mathbb{R}_+^*$, and if $\omega \in \mathbb{Z}$ is the fundamental frequency, this last should be scaled by $\omega' = \frac{\pi}{l} \omega$.

## E    PROOF OF THE THEOREM 3.19

*Proof.* Consider the function:

$$F(x) = \frac{\sqrt{2}}{n} \max_{k=0}^{n} \{1 + kx\} = \frac{\sqrt{2}}{n} \left( 1 + \max_{k=0}^{n} kx \right) \tag{70}$$

Note that since $x \in \mathbb{R}$, the maximum over $k$ depends on the sign of $x$:

- If $x > 0$, then $\max_{k=0}^{n} \{kx\} = nx$.
- If $x \leq 0$, then $\max_{k=0}^{n} \{kx\} = 0$ (achieved at $k = 0$).

Thus, we can write

$$F(x) = \begin{cases} \frac{\sqrt{2}}{n}(1+nx) = \sqrt{2}x + \frac{\sqrt{2}}{n}, & x > 0 \\ \frac{\sqrt{2}}{n}, & x \le 0 \end{cases} \tag{71}$$

We now analyze the variance of $F(x)$ under the assumption that $x \sim \mathcal{N}(0,1)$ (standard normal input):

As $n \to \infty$, the function becomes approximately:

$$F(x) \approx \begin{cases} \sqrt{2}x, & x > 0 \\ 0, & x \le 0 \end{cases} \tag{72}$$

This is similar to a scaled ReLU: $F(x) \approx \sqrt{2} \cdot \max(0, x)$, which we know from Appendix B has unitary forward and backward gains.

$\square$

# F  DEEP POLYNOMIALLY ACTIVATED NEURAL NETWORKS ARE MULTIVARIATE POLYNOMIAL MAPPINGS

Deep MLPs are compositions of affine transformations and activation functions applied layer by layer. When the activation functions are polynomial, the entire network can be expressed as a polynomial mapping.

**Definition F.1.** Let $n, m \in \mathbb{N}$. A function $F : \mathbb{R}^n \to \mathbb{R}^m$ is called a *polynomial mapping* if each component function $F_i : \mathbb{R}^n \to \mathbb{R}$, for $i = 1, \dots, m$, is a polynomial in $n$ variables. Explicitly, this means that for each $i$, $F_i$ has the form:

$$F_i(x_1, \dots, x_n) = \sum_{|\alpha| \le d_i} c_{i,\alpha} x_1^{\alpha_1} x_2^{\alpha_2} \cdots x_n^{\alpha_n}, \tag{73}$$

where the sum is taken over all multi-indices $\alpha = (\alpha_1, \dots, \alpha_n) \in \mathbb{N}^n$ such that $|\alpha| = \alpha_1 + \alpha_2 + \cdots + \alpha_n \le d_i$, $c_{i,\alpha} \in \mathbb{R}$ are real coefficients, and $d_i \in \mathbb{N}$.

**Definition F.2.** A *deep neural network* with $L$ layers, input dimension $n$, and output dimension $m$ is a function $F : \mathbb{R}^n \to \mathbb{R}^m$ of the form:

$$F(x) = W_L \sigma(W_{L-1} \sigma(\cdots \sigma(W_1 x + b_1) \cdots) + b_{L-1}) + b_L, \tag{74}$$

where $\forall i \in [\![1, L]\!]$ $C_i \in \mathbb{N}^*$. Each $W_i \in \mathbb{R}^{C_i \times C_{i-1}}$ is a weight matrix, $b_i \in \mathbb{R}^{C_i}$ is a bias vector, and $\sigma$ is an activation function applied element-wise.

**Proposition F.3.** *Let $F : \mathbb{R}^n \to \mathbb{R}^m$ be a deep neural network with polynomial activation functions of degree $d$. Then $F$ is a polynomial mapping of degree at most $d^L$. Furthermore, any $L$-layer MLP could be collapsed into an equivalent 3-layer network with the middle layer being a polynomial mapping of degree at most $d^L$.*

*Proof.* The proof proceeds by induction on the number of layers $L$ and is detailed in what follows.

**Base case:** For $L = 1$, the network takes the form

$$F(x) = W_1 \sigma(W_0 x + b_0) + b_1.$$

Since $\sigma$ is a polynomial of degree $d$, applying it to the affine transformation $W_0 x + b_0$ yields a polynomial mapping of degree at most $d$. Therefore, $F(x)$ is a polynomial mapping of degree at most $d$.

**Inductive step:** Assume the statement holds for $L - 1$ layers, meaning the network $F_{L-1}(x)$ is a polynomial mapping of degree at most $d^{L-1}$. For the $L$-layer case, we have

$$F(x) = W_L \sigma(F_{L-1}(x)) + b_L.$$

Since $\sigma$ is a polynomial of degree $d$, applying it to $F_{L-1}(x)$ results in a polynomial of degree at most $d \cdot d^{L-1} = d^L$. Thus, by induction, the statement holds for all $L \ge 1$. $\square$

**Corollary F.4.** *Any deep neural network with polynomial activation functions realizes a polynomial mapping whose degree grows exponentially with the number of layers.*

*Remark* F.5. The total number of monomial terms in this mapping is $\binom{d^L+n}{d^L}$.

*Remark* F.6. An equivalent consideration for trigonometric polynomials can be established by approximation, but will not be covered here.

## G  ALGORITHMS

---

**Algorithm 1** Initialization of Hermite Grid and Coefficients

---

**Input:** Polynomial degree $n$
**Output:** Coefficients tensor coeffs, Grid of powers tensor grid

Initialize coeffs and grid as zero matrices of shape $[n+1, n//2+1]$
**for** $i = 0$ **to** $n$ **do**
   **for** $j = 0$ **to** $\frac{n}{2}$ **do**
      **if** $j \leq \frac{i}{2}$ **then**
         $\text{coeffs}[i][j] \leftarrow (-1)^j e^{(-\log(j!)-\log((i-2j)!)-j\log(2))}$
         $\text{grid}[i][j] \leftarrow i - 2j$
      **else**
         $\text{coeffs}[i][j] \leftarrow 0$
         $\text{grid}[i][j] \leftarrow 0$
      **end if**
   **end for**
**end for**
**return** coeffs, grid

---

---

**Algorithm 2** Hermite Activation Function Forward Pass

---

**Input:** Input tensor $x$, polynomial degree $n$
**Parameters:** Learnable polynomial coefficients $A \in \mathbb{R}^n$
**Output:** Output tensor after applying Hermite activation function

$\text{coeffs}, \text{grid} \leftarrow \text{Initialize\_coeffs\_grid}()$
**Procedure** Forward($x$):
   $x \leftarrow x.\text{repeat}(n+1).\text{repeat}(n//2+1)$
   $x \leftarrow |x|^{\text{grid}} \odot \text{sign}(x)^{\text{grid}}$
   $x \leftarrow x@\text{coeffs}$
   $x \leftarrow x@A$
   **return** $x$
**End Procedure**

---

---

**Algorithm 3** Hermite Forward CUDA Kernel

---

**Input:** Input tensor $x$, degree $n$, output tensor out
**Output:** Computed Hermite polynomials up to degree $n$

**Procedure** HermiteForwardCUDA($x, n, \text{out}$):
   **for** $i$ in parallel index size($x$):
      $\text{out}[i \cdot n] \leftarrow 1.0$
      if $n > 1$: $\text{out}[i \cdot n + 1] \leftarrow x[i]$
      for $k = 2$ to $n$:
         $\text{out}[i \cdot n + k] \leftarrow x[i] \cdot \text{out}[i \cdot n + k - 1] - (k-1) \cdot \text{out}[i \cdot n + k - 2]$
**End Procedure**

---

---

**Algorithm 4** Hermite Backward CUDA Kernel

---

**Input:** Input tensor $x$, degree $n$, output tensor out, gradient tensor grad_out
**Output:** Computed gradients for Hermite polynomials

**Procedure** HermiteBackwardCUDA($x, n, $ out, grad_out):
    **for** $i$ in parallel index size(grad_out):
        grad $\leftarrow 0.0$
        **for** $k = 1$ to $n$:
            grad $\leftarrow$ grad $+ x[i \cdot n + k] \cdot k \cdot$ out$[i \cdot n + k - 1]$
        grad_out$[i] \leftarrow$ grad
**End Procedure**

---

**Algorithm 5** Fourier Activation Function Forward Pass

---

**Input:** Input tensor $x$, degree $n$
**Parameters:** Learnable coefficients $A \in \mathbb{R}^n$, fundamental $a \in \mathbb{R}$, phases $P \in \mathbb{R}^n$, frequencies $F \in \mathbb{R}^n$,
**Output:** Output tensor after applying Fourier activation function

**Procedure** FourierActivation($x$):
    $x \leftarrow x$.repeat($n + 1$)
    $x \leftarrow F \odot x - P$
    $x \leftarrow \sqrt{2}\cos(x)$
    $x \leftarrow x@A$
    $x \leftarrow x + a$
    **return** $x$
**End Procedure**

---

**Algorithm 6** Tropical Activation Function Forward Pass

---

**Input:** Input tensor $x$, degree $n$
**Parameters:** Learnable coefficients $A \in \mathbb{R}^n$
**Output:** Output tensor after applying Tropical activation function

powers $\leftarrow$ range(0,n+1)
**Procedure** Forward($x$):
    $x \leftarrow x$.repeat($n + 1$)
    $x \leftarrow \sqrt{2}/n \cdot \max(x \odot \text{powers} + A, \dim = -1)$
    **return** $x$
**End Procedure**

---

## H RATIONAL TROPICAL ACTIVATION

**Definition H.1.** The tropical quotient $\oslash$ of $x$ over $y$ is defined as:

$$x \oslash y := x - y \tag{75}$$

**Definition H.2.** The *tropical rational activation* $F$ is defined as the *quotient* of two tropical polynomials $F_1$ and $F_2$ of degree $m, n \in \mathbb{N}^2$ respectively.

$$F: \mathbb{R} \to \mathbb{R}$$
$$F(x) \mapsto F_1(x) \oslash F_2(x) := F_1(x) - F_2(x) \tag{76}$$

An example of fitting a classical activation (GELU) with a rational tropical activation is shown in Figure 5. Rational tropical activation is understood here in the general sense, i.e., with real powers.

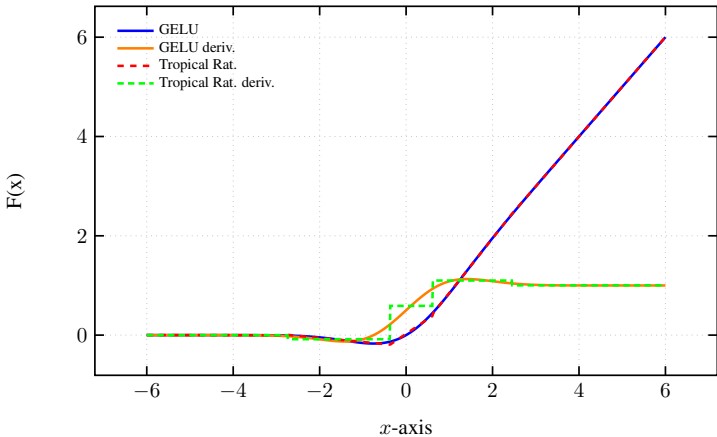

Figure 5: Hermite interpolation of a GELU with a Tropical Rational Activation of degree 6 in both the numerator and the denominator.

An example of fitting a convex function ($x \mapsto \frac{x^2}{2}$) with a polynomial tropical function (in the general sense) is shown in Figure 6.

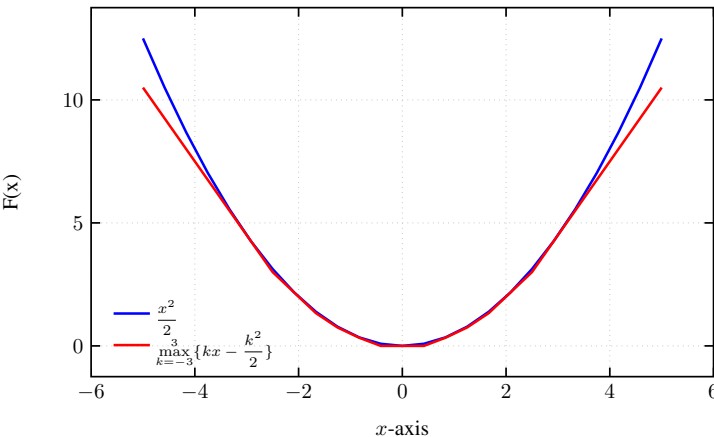

Figure 6: Interpolation of $\frac{x^2}{2}$ function by the Tropical-Laurent polynomial (with potentially negative powers) $\max\limits_{k=-3}^{3}\{kx - \frac{k^2}{2}\}$ of degree 6.

## I ABLATION STUDIES

Table 3: Ablation studies for the degree of the activation on ConvNeXt-T model.

| Activation | Degree | Train Loss | Val Top-1 (%) | Val Top-5 (%) |
|---|---|---|---|---|
| Tropical | 1 | 2.925 | 81.60 | 95.73 |
| Tropical | 3 | 2.866 | 82.01 | 95.91 |
| Tropical | 5 | 2.863 | 82.18 | 96.00 |
| Tropical | 6 | 2.857 | 82.20 | 95.90 |
| Fourier | 1 | 2.872 | 80.29 | 95.03 |
| Fourier | 3 | 2.850 | 80.61 | 95.26 |
| Fourier | 5 | 2.844 | 80.69 | 95.41 |
| Fourier | 6 | 2.837 | 80.93 | 95.44 |
| Hermite | 2 | 2.833 | 81.66 | 95.71 |
| Hermite | 3 | 2.790 | 82.34 | 96.03 |

Table 4: Ablation studies for the initialization of the activation on ConvNeXt-T model.

| Activation | Degree | Initialized from | Train Loss | Val Top-1 (%) | Val Top-5 (%) |
|---|---|---|---|---|---|
| Fourier | 6 | GELU | 2.775 | 81.91 | 95.77 |
| Fourier | 6 | Thrm. 3.13 | 2.837 | 80.93 | 95.44 |
| Hermite | 3 | GELU | 2.809 | 82.04 | 95.91 |
| Hermite | 3 | Thrm. 3.8 | 2.790 | 82.34 | 96.03 |

Table 5: Ablation studies for the learnability of the parameters of the activation on ConvNeXt-T model.

| Activation | Degree | Learnable? | Train Loss | Val Top-1 (%) | Val Top-5 (%) |
|---|---|---|---|---|---|
| Tropical | 6 | × | 3.560 | 76.31 | 93.09 |
| Tropical | 6 | √ | 2.857 | 82.20 | 95.90 |
| Fourier | 6 | × | 3.181 | 79.51 | 94.60 |
| Fourier | 6 | √ | 2.837 | 80.93 | 95.44 |
| Hermite | 3 | × | 3.411 | 78.48 | 94.20 |
| Hermite | 3 | √ | 2.790 | 82.34 | 96.03 |

Table 6: Ablation studies for the clamping in the Hermite activation on ConvNeXt-T model.

| Activation | Degree | Clamped? | Train Loss | Val Top-1 (%) | Val Top-5 (%) |
|---|---|---|---|---|---|
| Hermite | 3 | √ | 2.772 | 81.98 | 95.81 |
| Hermite | 3 | × | 2.790 | 82.34 | 96.03 |

## J A BRIEF DIGRESSION ON KOLMOGOROV ARNOLD NETWORKS (KANS)

Kolmogorov-Arnold networks Liu et al. (2025) have been presented as a potential alternative to Multilayer-Perceptrons (MLPs), promoting several merits such as greater accuracy, fewer learnable parameters, and better interpretability. While the first two advantages could only be demonstrated for simple cases in the (Liu et al., 2025) article, the third benefit is more straightforward, as these networks overcome the "black-box" aspect of traditional non-linear activations MLPs by allowing

the activation to be polynomial, piece-wise polynomial, or rational, as in Yang & Wang (2025). From there, having learned the weights of the network and those of the activation, it becomes clear what approximation these functions (polynomial, rational, or trigonometric ) have converged to.

Rather than providing a direct application of the celebrated Kolmogorov-Arnold representation theorem (KART) Kolmogorov (1957); Arnold (1959), the recent work on KAN Liu et al. (2025) appears to take inspiration from it in a more figurative sense. For clarity, we recall that the Kolmogorov-Arnold representation theorem, cited below, states that any continuous multivariate function $f : [0, 1]^n \to \mathbb{R}$ can be represented as a composition of addition and some functions of one variable denoted by $\psi_{q,p}$ and $\Phi_q$:

**Theorem J.1.** *(Arnold (2009b;a)) Let $f : \mathbb{I}^n := [0, 1]^n \to \mathbb{R}$ be an arbitrary multivariate continuous function. Then it can be represented as follows:*

$$f(x_1, \ldots, x_n) = \sum_{q=0}^{2n} \Phi_q \left( \sum_{p=1}^{n} \psi_{qp}(x_p) \right) \tag{77}$$

*with continuous one-dimensional functions $\Phi_q \colon \mathbb{R} \to \mathbb{R}$ and $\psi_{q,p} \colon [0, 1] \to \mathbb{R}$. $\Phi_q$ are called outer funcions and $\psi_{q,p}$ are called inner functions. The inner functions $\psi_{q,p}$ are independent of the function $f$.*

This differs substantially from KAN's formulation Liu et al. (2025), where the outer functions disappear, the inner functions are replaced by a weighted sum of a SiLU MLP Elfwing et al. (2018) and a B-spline, and the networks are a composition of multiple feed-forward layers to accommodate recent neural network architectures.

Since the KART proof is not constructible, and is essentially based on Baire's theorem Kahane (1975), the first efforts to implement a constructive proof of the KART were made by Sprecher in Sprecher (1996; 1997). These latest works are based on a more economical variant of the KART in terms of the number of outer and inner functions due to both Sprecher (1965) and Lorentz (1966).

This was followed by the first article on the practical training of this type of network by Köppen (2002), pointing out at the same time that the inner function $\psi$ constructed in this theorem was continuous but fractal! This limited its use in gradient-based learning algorithms. Braun & Griebel (2009) gave rigorous proof of termination, continuity, and monotonicity for the construction of the inner and the outer functions given by Sprecher (1997).

As acknowledged by both Liu et al. (2025) and Yang & Wang (2025), the original "KAN" layer defined in Liu et al. (2025) could be seen as a sum of a SiLU MLP and a weighted B-Spline combination. Let us define a linear function $\mathcal{L}_W \colon x \mapsto Wx$, with $W$ a learnable weight matrix. The "KAN" layer Liu et al. (2025) is then defined as follows:

$$\texttt{KAN}_{\texttt{Liu}}(x) = \mathcal{L}_{W_b}(\texttt{SiLU}(x)) + \mathcal{L}_{W_s}\left(\sum_i c_i B_i(x)\right) \tag{78}$$

With $W_b$ and $W_s$ two learnable weight matrices, $(B_i)_{i \in [\![0,d]\!]}$ a family of B-spline functions of order $d + 1$, $(c_i)_{i \in [\![0,d]\!]}$ the learnable spline weights and $\texttt{SiLU} \colon x \mapsto \frac{x}{1+e^{-x}}$.

Indeed, if we follow the line of thought set out in KAN Liu et al. (2025), an MLP with learnable activation, or equivalently a learnable activation network (LAN) would be a sort of KART formulation, with the $\psi_{qp}$ inner functions being a linear combination of ReLU functions. However, this is not what the KART theorem suggests. Constructing a Kolmogorov-Arnold superposition requires a maximum of two layers formulated by inner and outer functions as in theorem J.1 (Ismailov, 2025).

It is worth noting that the concept of using splines to approximate inner functions in a Kolmogorov-Arnold network or more generally as a representation of an activation function isn't entirely new. The analogy between KANs and MLPs has been noticed since Hecht-Nielsen (1987) and Kůrková (1992). Earlier research, such as Igelnik & Parikh (2003), introduced Kolmogorov's Spline Network, which employed splines for flexible function approximation. In his PhD thesis, Braun (2009) corrected the constructive proof of the KAT and gave practical examples using B-splines. Further developments in this area include Bohra et al. (2020) and Fakhoury et al. (2022), who focused on learning adaptive activation functions through splines, thus enhancing the network's expressiveness.

Additionally, the use of the Kolmogorov superposition theorem to tackle high-dimensional problems has been explored by Laczkovich (2021) and Lai & Shen (2021), who showed its potential in overcoming the curse of dimensionality. Similarly, Montanelli et al. (2021) demonstrated how structured networks like Deep ReLU models can efficiently approximate bandlimited functions, thus expanding the practical applications of spline-based methodologies in neural networks.

With an equivalent number of parameters or FLOP, Yu et al. (2024) observed that KAN surpasses MLP solely in symbolic formula representation, while it falls short of MLP in other machine learning tasks, including computer vision, NLP, and audio processing. Cang et al. (2024) confirmed the same finding.

Nevertheless, KANs have had the merit of rekindling interest in learnable activations in neural networks, among them polynomial and trigonometric activations.

Since the interest in KANs began, numerous researchers have proposed a multitude of learnable functions for activations, spanning a diverse range of mathematical functions, including splines, classical orthogonal polynomials, rational functions, Fourier bases, and wavelets... Despite this, in some instances, the safety of these operations, the boundedness of their gradients, their initialization, and their computational properties in the context of gradient descent have sometimes received less emphasis. Instead, many studies have highlighted proof-of-concept results, often demonstrating that such functions can achieve strong performance on benchmark datasets like MNIST LeCun et al. (1998). This line of work has produced a rich body of literature. A common observation, however, is that much of it focuses on adapting a specific interpolation function within relatively shallow architectures and evaluating on small-scale datasets (such as the MNIST dataset, for example). Because a wide variety of functions can achieve test accuracies exceeding 97% on MNIST with networks of depth three or less, it becomes challenging to distinguish which approaches provide the most robust or generalizable benefits.

## K    EXTENDED RELATED WORK

The subject of learnable activation is a well-known one, but it has seen a resurgence thanks to the popularity enjoyed by the KAN article Liu et al. (2025). Examples of works in which the main theme is learning the activation function include Houlsby et al. (2019); Goyal et al. (2020); Tavakoli et al. (2021); Moosavi et al. (2022); Fang et al. (2023); Bodyanskiy & Kostiuk (2023); Pishchik (2023).

The use of orthogonal polynomials as activation functions predates many recent developments. For example, Ma & Khorasani (2005) introduced Hermite polynomial activations within a constructive 1-layer neural network framework, while Venkatappareddy et al. (2021) explored Legendre polynomial-based activations to better approximate max-pooling behavior.

Earlier works exploring polynomial activations in deep neural networks trained using the backpropagation algorithm include Zhou et al. (2019) and Chrysos et al. (2020), which empirically demonstrate that polynomially activated neural networks, even without non-linear activation functions, can perform well across multiple tasks. Building on this, Chrysos et al. (2023) sought to regularize such networks to compete with deep ReLU networks.

More recently, Nebioglu & Iliev (2023) investigated the use of Chebyshev and Hermite orthogonal polynomials as activation functions, demonstrating that Chebyshev activations are computationally efficient but sensitive to problem types, while Hermite activations exhibit greater robustness and generalization. Additionally, Xiao et al. (2024) introduced HOPE (High-order Polynomial Expansion), a novel method that represents neural networks as high-order Taylor polynomials, enabling improved interpretability, low computational complexity, and applications such as function discovery, fast inference, and feature selection.

Other recent works utilizing Chebyshev activation include Deepthi et al. (2023) and Rezaeian et al. (2025), which employed single-layer shallow networks. Seydi (2024) conducted a comparative study of exotic polynomial activations on the MNIST dataset, while Cooley et al. (2024) applied polynomial-augmented neural networks for approximating solutions to partial differential equations.

On the rational activation front, notable works include Trefethen & Gutknecht (1987), which introduced stable-Padé and Chebyshev-Padé approximators, and Molina et al. (2020), which proposed the Safe-Padé activation by ensuring the denominator of the rational activation remains nonzero. An

orthogonal variant of the Padé approximant was presented in Biswas et al. (2021), while Chebyshev rational functions Castellanos & Rosenthal (1993) and Fourier rational functions Geer (1995) were explored in subsequent studies. More recently, advancements in rational activation using general Jacobi functions were introduced in Aghaei (2024b;a).

Polynomial piecewise functions (such as B-splines) and rational functions (such as the Padé approximant) can exhibit finite support properties. On the other hand, these last lack the orthogonality property. Several works have aimed to formulate orthogonal splines Mason et al. (1993); Alavi & Aminikhah (2023) and orthogonal rational functions Bultheel et al. (2001), or even a theory of spline wavelets Chui & Wang (1991) and rational wavelets Zheng & Minggen (1999); Choueiter & Glass (2007).

Learning with a periodic function or a Fourier series has also been the subject of many anterior works, such as Sitzmann et al. (2020), and more recently Mehrabian et al. (2025), and Martinez-Gost et al. (2024) using a Discrete Cosine Transform (DCT). Recently, Hashemi et al. (2025) introduced the Dynamic Range Activator (DRA), an activation function that combines harmonic (trigonometric) and hyperbolic components to capture the highly recursive and high-variance behavior within a deep problem in enumerative algebraic geometry.

In the context of tropical activations, prior work has been done to establish connections between tropical geometry and neural networks. For instance, Zhang et al. (2018) demonstrated that feedforward neural networks with ReLU activation can be interpreted as tropical rational maps, relating their decision boundaries to tropical hypersurfaces and showing how deeper networks leverage zonotopes to achieve exponentially greater expressiveness. Building on this geometric foundation, Smyrnis & Maragos (2019) introduced tropical polynomial division, an approach inspired by the max-plus semiring, and applied it to neural networks with ReLU activation. Recent work also developed tropical activation functions Yoshida et al. (2024), which were subsequently applied to convolutional neural networks (CNNs) for image classification tasks on MNIST LeCun et al. (1998), CIFAR10 Krizhevsky et al. (2009), and SVHN Netzer et al. (2011) in Pasque et al. (2026).

The stability of deep networks with learnable or unconventional activations is closely tied to variance-preserving initialization. Prior analyses have established theoretical conditions for stable signal propagation across depth Hanin & Rolnick (2018), while empirical and theoretical studies have examined variance decay phenomena in ReLU networks Luther & Seung (2019). For sinusoidal activations, Passalis et al. (2019) introduced a dedicated variance-preserving initialization scheme.

## L IMAGE CLASSIFICATION RESULTS ON CIFAR10

We conducted an experiment using ConvNeXt-T on CIFAR10 for 300 epochs and averaged the results over 10 different seeds. The experiment shows that Hermite, Fourier, and Tropical activations are significantly above the GELU baseline in terms of test metrics. In addition, we added three different seeds for ResNet50 with ReLU, and the results of the latter are clearly inferior to those of ConvNeXt-T, ConvNeXt-T being a modernized version of ResNet50. The results, in table form, are reported in Table 7 and in graphical form in Figures 7, 8, and 9.

Table 7: Comparison of the proposed activation functions on ResNet50 and ConvNeXt-T models for CIFAR-10 image classification task. Values are reported as mean ± standard deviation over 10 different seeds (3 seeds for the ResNet50 case). p-values (two-tailed Student's t-test assuming equal variances) compare each activation's Top1-accuracy against GELU.

| Model | Activation | Top-1 Acc. (%) | Top-5 Acc. (%) | p-value vs GELU (Top-1) |
|---|---|---|---|---|
| ResNet50 | **ReLU** | $88.9 \pm 0.04$ | $99.43 \pm 0.47$ | $< 0.0001$ (****) |
| ConvNeXt-Tiny | **Baseline (GELU)** | $90.47 \pm 0.20$ | $99.62 \pm 0.06$ | – |
| ConvNeXt-Tiny | **Tropical** | $90.87 \pm 0.19$ | $99.63 \pm 0.04$ | $0.0002$ (***) |
| ConvNeXt-Tiny | **Fourier** | $91.23 \pm 0.65$ | $99.60 \pm 0.05$ | $0.0023$ (**) |
| ConvNeXt-Tiny | **Hermite** | $\mathbf{91.35} \pm 0.29$ | $99.63 \pm 0.05$ | $< 0.0001$ (****) |

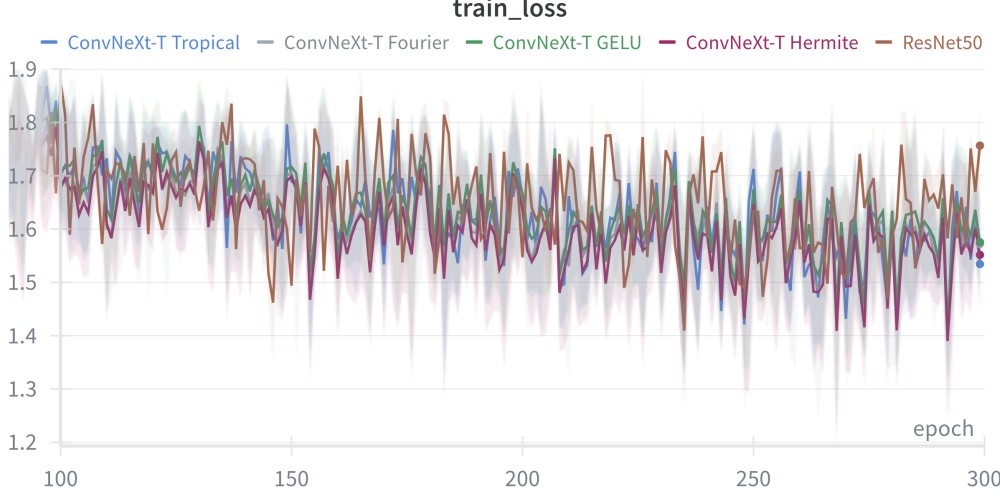

Figure 7: Training loss curves for ConvNeXt-T on CIFAR-10. The solid lines represent the mean of the metric over 10 different seeds (3 seeds for the ResNet50 case), and the shaded areas show the range (min to max).

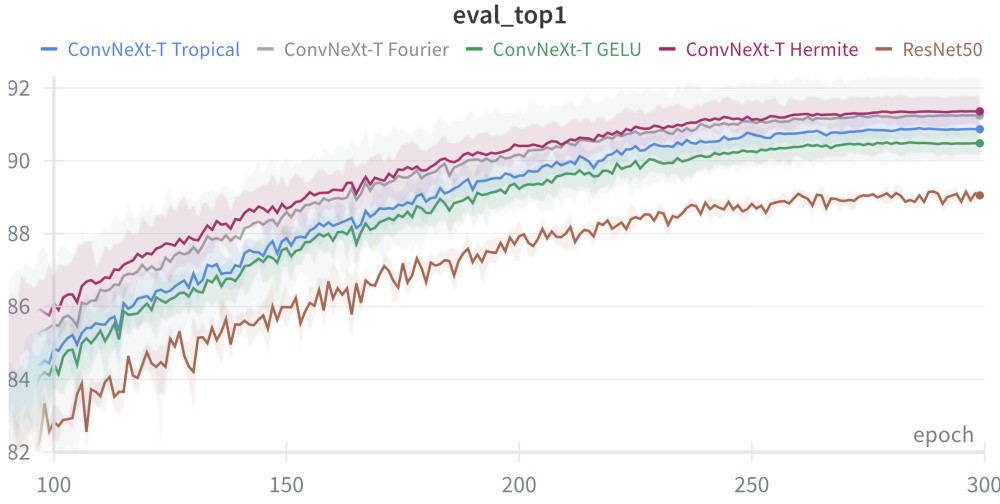

Figure 8: Evaluation Top1 accuracy curves for ConvNeXt-T on CIFAR-10. The solid lines represent the mean of the metric over 10 different seeds (3 seeds for the ResNet50 case), and the shaded areas show the range (min to max).

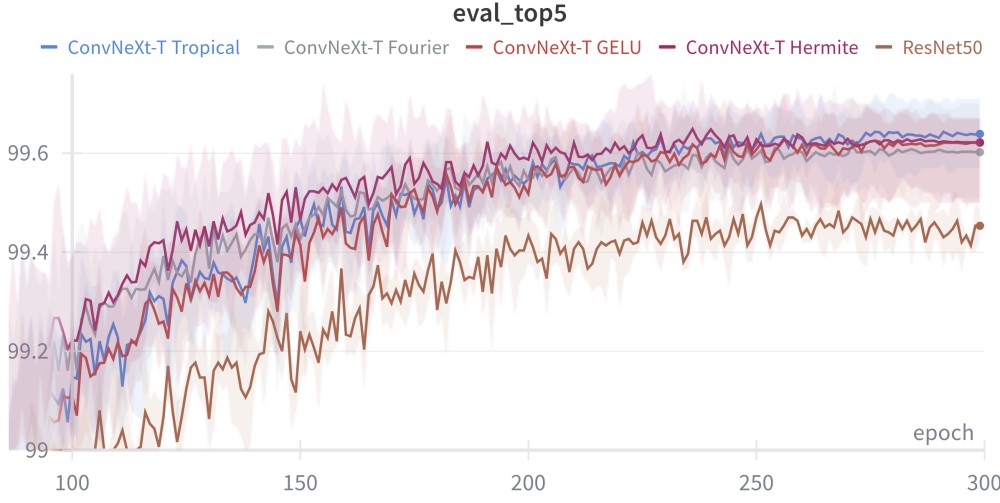

Figure 9: Evaluation Top5 accuracy curves for ConvNeXt-T on CIFAR-10. The solid lines represent the mean of the metric over 10 different seeds (3 seeds for the ResNet50 case), and the shaded areas show the range (min to max).

For this experiment we used the TIMM library Wightman (2019) with the following configuration 8:

Table 8: Configuration for the pretraining experiment on CIFAR-10.

| Hyperparameter | Value |
| --- | --- |
| Input Size | 3×32×32 |
| Number of Classes | 10 |
| Batch Size | 128 × 4 GPUs |
| Optimizer | AdamW |
| Learning Rate | 4e-3 |
| Epochs | 300 |
| Scheduler | Cosine |
| Drop Path Rate | 0.0 |
| Mean | 0.491, 0.482, 0.446 |
| Std | 0.247, 0.243, 0.261 |
| Warmup Epochs | 20 |
| Weight Decay | 0.0 |
| Mixup | 0.8 |
| Label Smoothing | 0.1 |
| Auto Augmentation | rand-m9-mstd0.5 |
| Re-mode | Pixel |
| Random Erasing Prob | 0.25 |
| Gradient Clipping | 5.0 |
| CutMix | 1.0 |

## M    DECISION BOUNDARIES

To evaluate how the choice of activation function affects classification behavior and boundary smoothness, we compare the decision boundaries of four single-layer neural networks trained on simple 2D classification datasets (e.g., two moons, circles, etc.). Each network uses a different activation function. The Hermite activation function produces a globally smooth and coherent decision boundary, reflecting its polynomial nature. Notably, when the regions of the classes intersect, the boundary can develop a critical point corresponding to the intersection, where the local geometry degenerates. The first sub-figure in the top left shows that the decision boundary is rather a plane curve resembling an elliptic curve with two connected components and no singular points. This curve delimits three regions of the 2D space. The network uses two of these regions to classify/cluster one of the two classes of points, while the complementary region classifies the points belonging to the second class. In contrast, the Fourier activation leads to quasi-periodic patterns that adapt well to the underlying structure of the data, capturing fine-grained details and even fitting noisy points. The tropical activation yields a piecewise affine boundary, resembling the behavior of ReLU, with sharp transitions and linear segments that reflect its max-plus (tropical) structure.

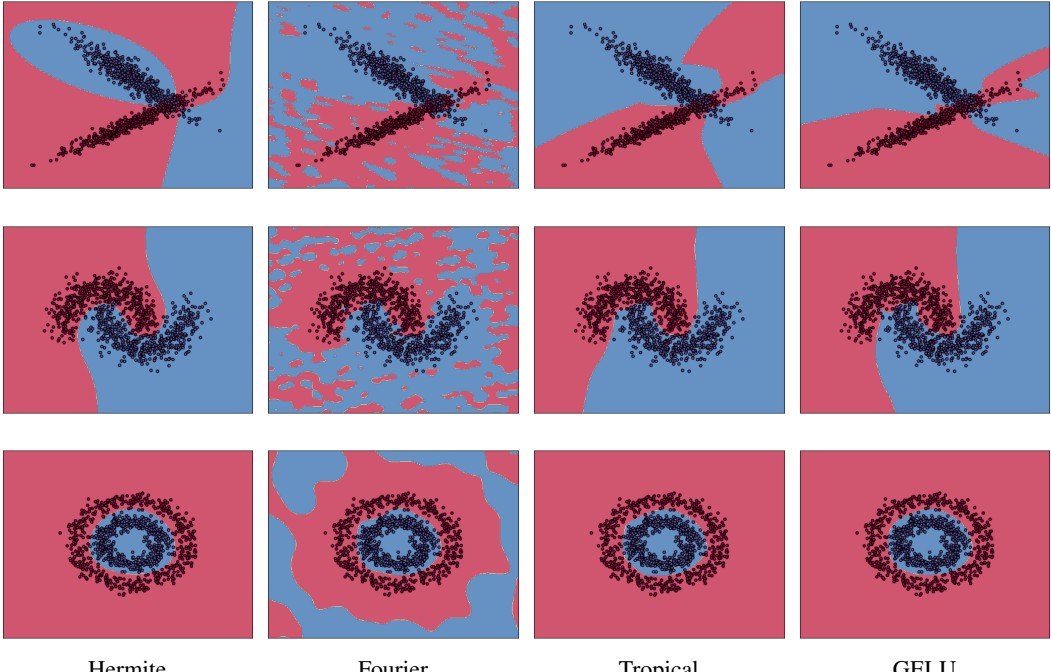

| Hermite | Fourier | Tropical | GELU |

Figure 10: Decision boundaries across datasets: Top row: classification; middle row: moons; bottom row: circles, using four different activation functions.

# N  LINE PLOTS

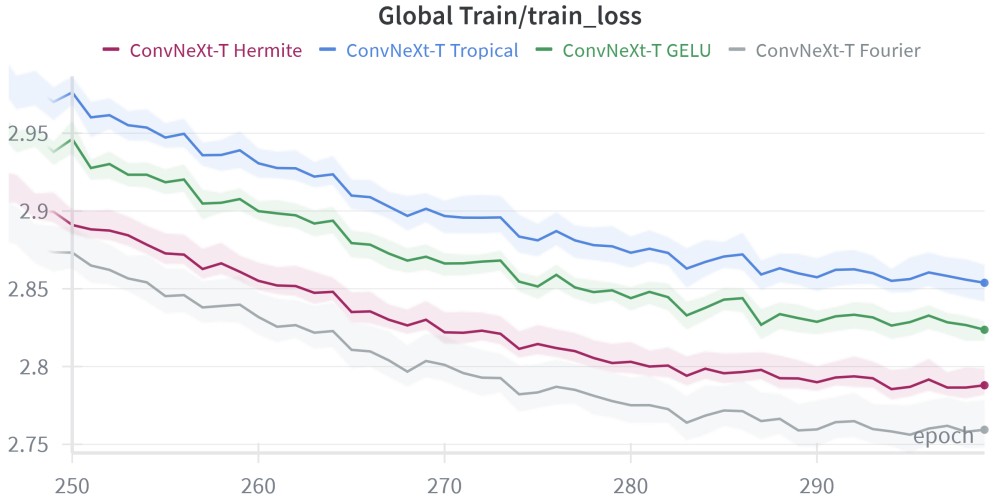

Figure 11: Training loss curves for ConvNeXt-T on ImageNet1k. The solid lines represent the mean of the metric over 5 different seeds, and the shaded areas show the range (min to max).

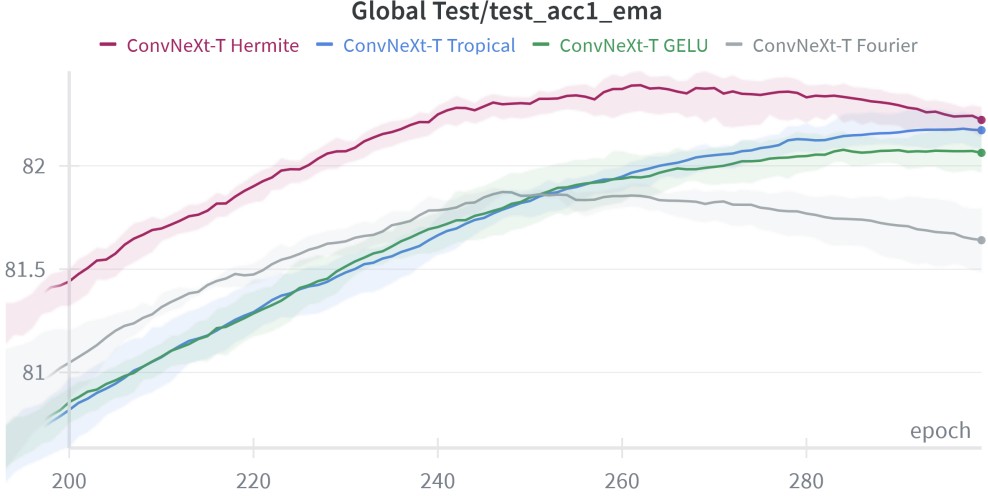

Figure 12: Top1 evaluation accuracy for ConvNeXt-T on ImageNet1k. The solid lines represent the mean of the metric over 5 different seeds, and the shaded areas show the range (min to max).

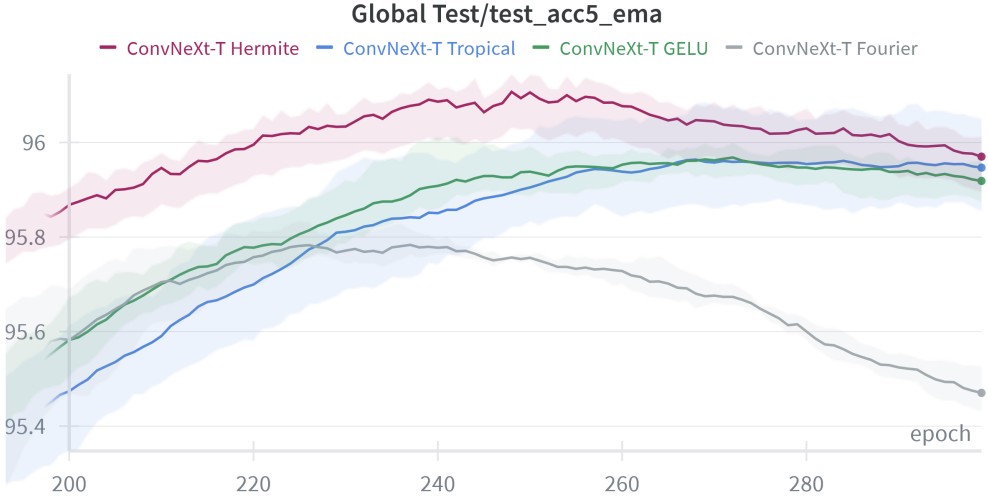

Figure 13: Top5 evaluation accuracy for ConvNeXt-T on ImageNet1k. The solid lines represent the mean of the metric over 5 different seeds, and the shaded areas show the range (min to max).

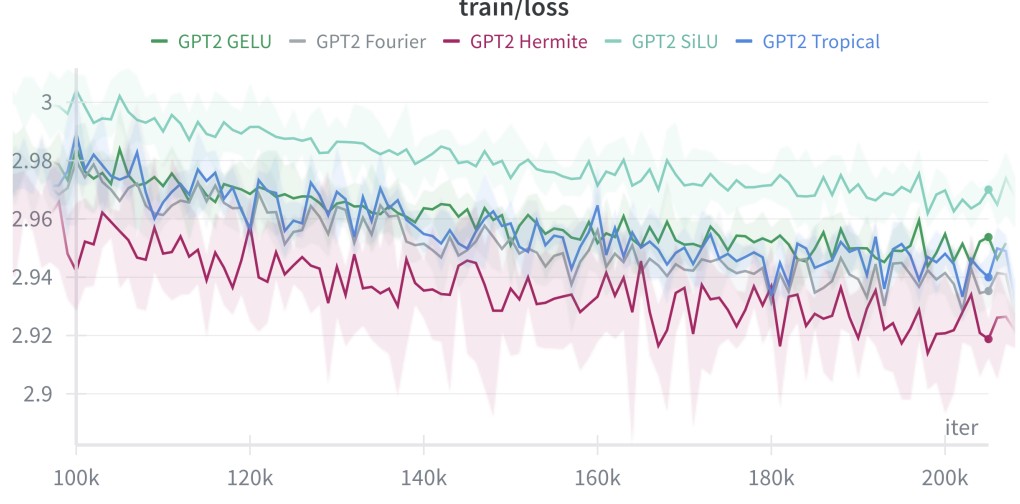

Figure 14: Comparison of the train losses of the GPT2 model (124M) on OpenWebText with GELU, SiLU, Hermite, Fourier, and Tropical activations. The solid lines represent the mean of the metric over 5 different seeds, and the shaded areas show the range (min to max).

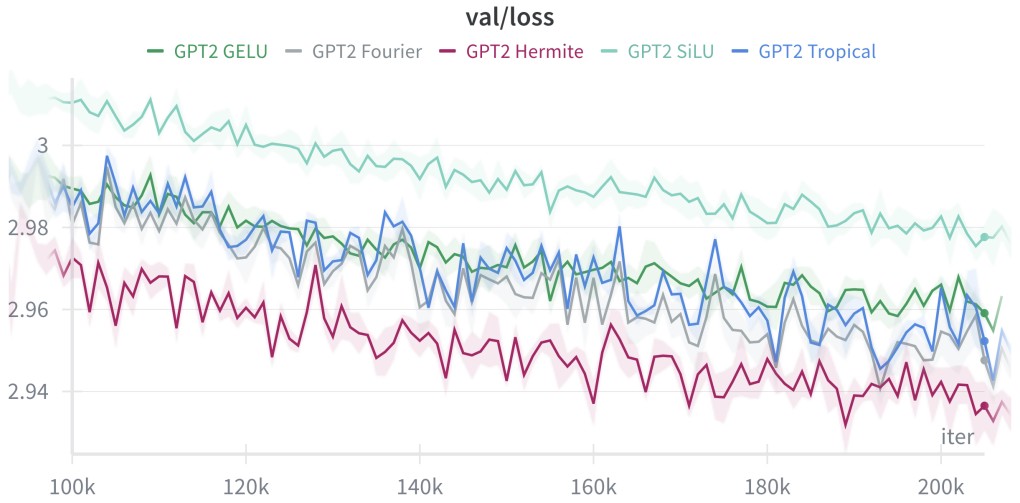

Figure 15: Comparison of the validation losses of the GPT2 model (124M) on OpenWebText with GELU, SiLU, Hermite, Fourier, and Tropical activations. The solid lines represent the mean of the metric over 5 different seeds, and the shaded areas show the range (min to max).

## O    FINETUNING ACTIVATIONS EXPERIMENT ON CIFAR10

We conducted an experiment for fine-tuning ConvNeXt-tiny (pre-trained on ImageNet1k) on CIFAR10. We froze all the weights except those of the last linear layer and the ones of the learnable activations, which were initialized by fitting GELU with a Hermite interpolation. The results hereby show a clear superiority of the proposed learnable activations:

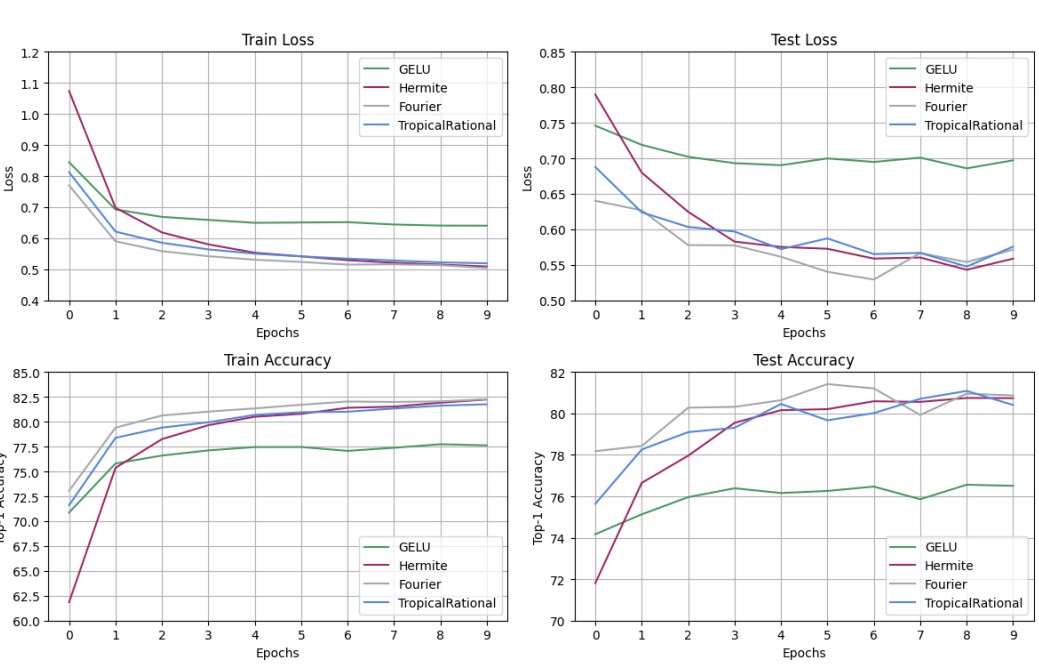

Figure 16: Performance of a pretrained ConvNeXt-T (on ImageNet1k) on CIFAR10 when fine-tuning only the final linear layer and the learnable coefficients of the activations.

## P TIMING RESULTS

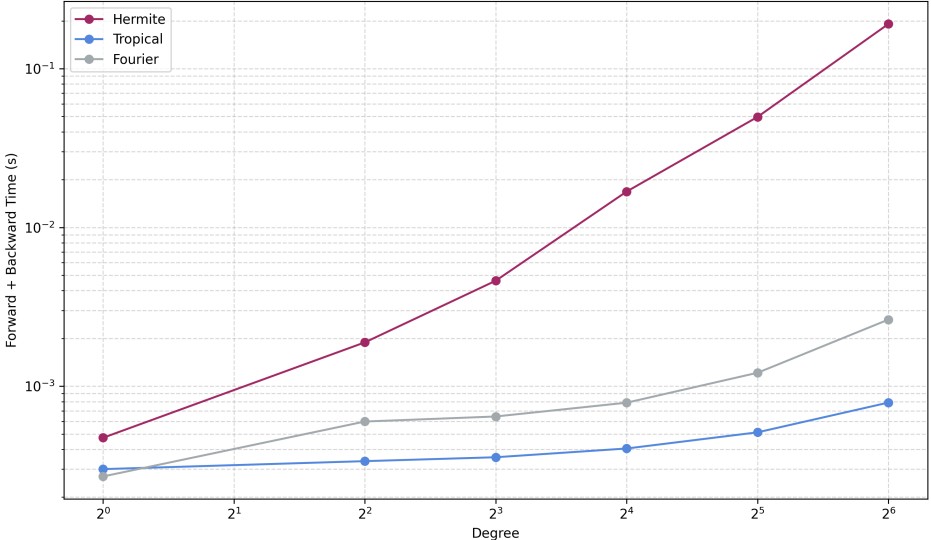

Figure 17: Forward + backward pass averaged times (in seconds and log-log scale) for Hermite, Tropical, and Fourier activations across varying degrees as benchmarked on an AMD EPYC 7402 CPU.

Table 9: Forward + backward pass averaged times (in seconds) for Hermite, Tropical, and Fourier activations across varying degrees as benchmarked on an AMD EPYC 7402 CPU.

| Activation | Degree | Forward+Backward Time |
|---|---|---|
| Hermite | 1 | 0.00047354 |
| Hermite | 4 | 0.00188883 |
| Hermite | 8 | 0.00462004 |
| Hermite | 16 | 0.0168367 |
| Hermite | 32 | 0.0496985 |
| Hermite | 64 | 0.191425 |
| Tropical | 1 | 0.000300329 |
| Tropical | 4 | 0.000337174 |
| Tropical | 8 | 0.000356829 |
| Tropical | 16 | 0.000405076 |
| Tropical | 32 | 0.000512147 |
| Tropical | 64 | 0.000788548 |
| Fourier | 1 | 0.000270138 |
| Fourier | 4 | 0.000599022 |
| Fourier | 8 | 0.000644515 |
| Fourier | 16 | 0.000787303 |
| Fourier | 32 | 0.00121512 |
| Fourier | 64 | 0.00262779 |

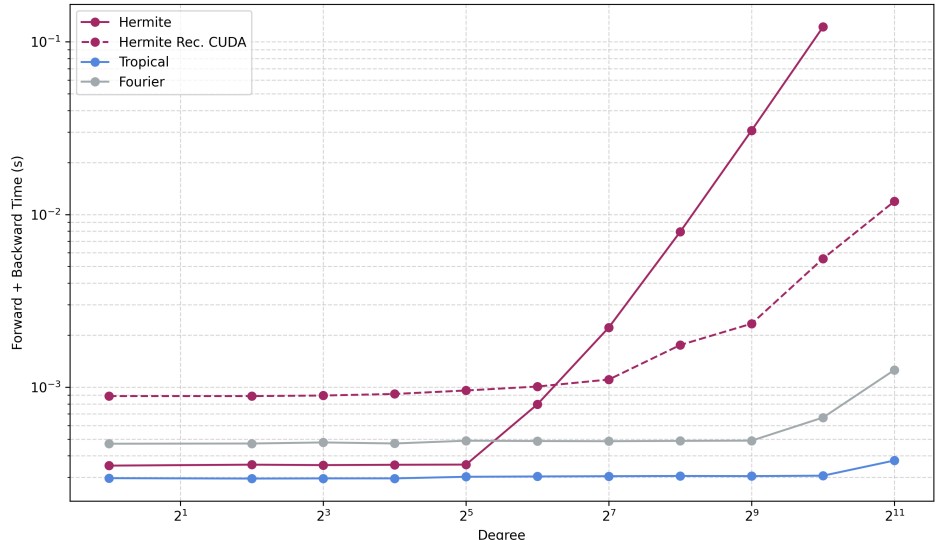

Figure 18: Forward + backward pass averaged times (in seconds) for Hermite (explicit Alg. 2, Eq. 22 and recurrence-based CUDA implementation Alg. 3, Eq. 23), Tropical, and Fourier activations across varying degrees as benchmarked on a single NVIDIA A100 GPU/40GB.

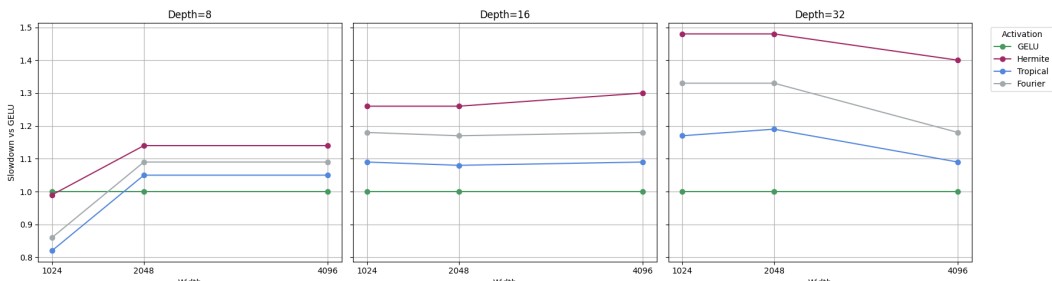

Figure 19: Relative slowdowns of Hermite, Tropical, and Fourier activations compared to GELU across different widths.

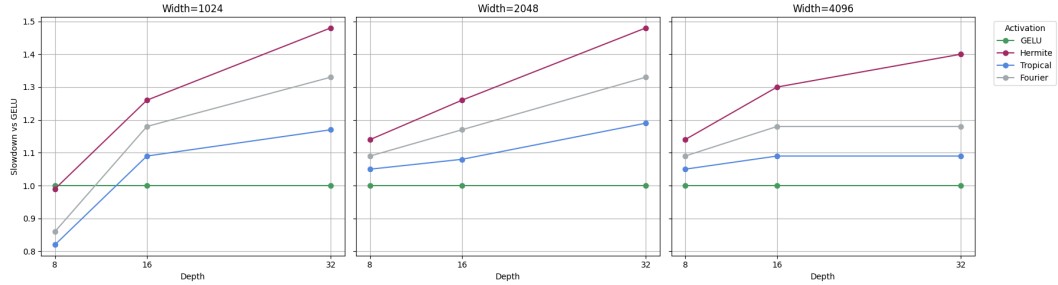

Figure 20: Relative slowdowns of Hermite, Tropical, and Fourier activations compared to GELU across different depths.

Table 10: Forward + backward pass averaged times (in seconds) for Hermite (explicit Alg. 2, Eq. 22 and recurrence-based CUDA implementation Alg. 3, Eq. 23), Tropical, and Fourier activations across varying degrees as benchmarked on a single NVIDIA A100 GPU/40GB.

| Activation | Degree | Forward+Backward Time |
|---|---|---|
| Hermite | 1 | 0.000350997 |
| Hermite | 4 | 0.00035552 |
| Hermite | 8 | 0.00035341 |
| Hermite | 16 | 0.000355005 |
| Hermite | 32 | 0.000355887 |
| Hermite | 64 | 0.000795927 |
| Hermite | 128 | 0.00221812 |
| Hermite | 256 | 0.00794526 |
| Hermite | 512 | 0.0306861 |
| Hermite | 1024 | 0.122032 |
| Hermite Rec. CUDA | 1 | 0.000887356 |
| Hermite Rec. CUDA | 4 | 0.000887024 |
| Hermite Rec. CUDA | 8 | 0.000893834 |
| Hermite Rec. CUDA | 16 | 0.000912833 |
| Hermite Rec. CUDA | 32 | 0.000956004 |
| Hermite Rec. CUDA | 64 | 0.00100782 |
| Hermite Rec. CUDA | 128 | 0.00110659 |
| Hermite Rec. CUDA | 256 | 0.0017534 |
| Hermite Rec. CUDA | 512 | 0.00233119 |
| Hermite Rec. CUDA | 1024 | 0.00555244 |
| Hermite Rec. CUDA | 2048 | 0.0119172 |
| Tropical | 1 | 0.000296884 |
| Tropical | 4 | 0.000295298 |
| Tropical | 8 | 0.000295942 |
| Tropical | 16 | 0.000296063 |
| Tropical | 32 | 0.000302484 |
| Tropical | 64 | 0.000303783 |
| Tropical | 128 | 0.000304883 |
| Tropical | 256 | 0.000305769 |
| Tropical | 512 | 0.000305247 |
| Tropical | 1024 | 0.000306931 |
| Tropical | 2048 | 0.000375793 |
| Fourier | 1 | 0.000470023 |
| Fourier | 4 | 0.000471404 |
| Fourier | 8 | 0.000478096 |
| Fourier | 16 | 0.000471656 |
| Fourier | 32 | 0.000489211 |
| Fourier | 64 | 0.000487039 |
| Fourier | 128 | 0.000486042 |
| Fourier | 256 | 0.00048811 |
| Fourier | 512 | 0.000489757 |
| Fourier | 1024 | 0.000666652 |
| Fourier | 2048 | 0.00125729 |

Table 11: Training times (in seconds) and relative slowdowns compared to GELU across different MLP network widths and depths, for Hermite, Tropical, and Fourier activations. The reported times were averaged per epoch and were obtained using a single NVIDIA A100 GPU with 40 GB of memory.

| Activation | Degree | Width | Depth | Training Time | Slowdown vs Baseline |
|---|---|---|---|---|---|
| GELU | - | 1024 | 8 | 16.13s | 1.00× |
| Hermite | 3 | 1024 | 8 | 16.03s | 0.99× |
| Tropical | 6 | 1024 | 8 | 13.27s | 0.82× |
| Fourier | 6 | 1024 | 8 | 13.93s | 0.86× |
| GELU | - | 1024 | 16 | 12.86s | 1.00× |
| Hermite | 3 | 1024 | 16 | 16.15s | 1.26× |
| Tropical | 6 | 1024 | 16 | 14.06s | 1.09× |
| Fourier | 6 | 1024 | 16 | 15.13s | 1.18× |
| GELU | - | 1024 | 32 | 13.96s | 1.00× |
| Hermite | 3 | 1024 | 32 | 20.65s | 1.48× |
| Tropical | 6 | 1024 | 32 | 16.29s | 1.17× |
| Fourier | 6 | 1024 | 32 | 18.50s | 1.33× |
| GELU | - | 2048 | 8 | 12.36s | 1.00× |
| Hermite | 3 | 2048 | 8 | 14.08s | 1.14× |
| Tropical | 6 | 2048 | 8 | 12.98s | 1.05× |
| Fourier | 6 | 2048 | 8 | 13.51s | 1.09× |
| GELU | - | 2048 | 16 | 12.96s | 1.00× |
| Hermite | 3 | 2048 | 16 | 16.29s | 1.26× |
| Tropical | 6 | 2048 | 16 | 14.05s | 1.08× |
| Fourier | 6 | 2048 | 16 | 15.15s | 1.17× |
| GELU | - | 2048 | 32 | 13.98s | 1.00× |
| Hermite | 3 | 2048 | 32 | 20.65s | 1.48× |
| Tropical | 6 | 2048 | 32 | 16.64s | 1.19× |
| Fourier | 6 | 2048 | 32 | 18.53s | 1.33× |
| GELU | - | 4096 | 8 | 12.43s | 1.00× |
| Hermite | 3 | 4096 | 8 | 14.12s | 1.14× |
| Tropical | 6 | 4096 | 8 | 13.02s | 1.05× |
| Fourier | 6 | 4096 | 8 | 13.58s | 1.09× |
| GELU | - | 4096 | 16 | 13.10s | 1.00× |
| Hermite | 3 | 4096 | 16 | 17.07s | 1.30× |
| Tropical | 6 | 4096 | 16 | 14.30s | 1.09× |
| Fourier | 6 | 4096 | 16 | 15.42s | 1.18× |
| GELU | - | 4096 | 32 | 23.93s | 1.00× |
| Hermite | 3 | 4096 | 32 | 33.41s | 1.40× |
| Tropical | 6 | 4096 | 32 | 26.10s | 1.09× |
| Fourier | 6 | 4096 | 32 | 28.21s | 1.18× |

## Q  LARGE LANGUAGE MODEL USAGE DISCLOSURE

We used large language models to assist in translating, rewording, and polishing the text for clarity and readability. The models were not used for idea generation, experiments, analysis, or contributions at the level of scientific authorship.

