# OpenReview forum: "Polynomial, trigonometric, and tropical activations"
_ICLR.cc/2026/Conference — ICLR 2026 Poster_

### Official Review · Reviewer_3rgG · 2025-10-28

**Soundness:** 4
**Presentation:** 3
**Contribution:** 3
**Rating:** 4
**Confidence:** 3

**Summary:**

This work reopens the discussion on which functions can be used in deep neural networks (or, more specifically, whether a few of the highlighted functions in this paper can). The paper covers theory and some empirical results. Chiefly, perhaps, is empirical work showing ConvNeXt and GPT-2 can be trained using orthogonal learnable activations, at least on specific shown datasets, which "eliminates the need for additional mechanisms".

**Strengths:**

- the theoretical underpinnings of Sec 3 are good -- 3.1 leads naturally to 3.2, 3.3, and 3.4. I may have missed _why_ Hermite, Fourier, Tropic activations are focused on. These families are unified cleanly.
- The practical implementation in Sec 3.5 seems like it's reproducible. At least, no obvious flaws with that secion was found.

**Weaknesses:**

- It's not *technically* a weakness, but convention seems to be moving away from older datasets like CIFAR-10, and older models like GPT-2. This does not sway the decision much, but it seems to upset 'the community' and it does limit the generalizability of your claims, especially to larger scales.
- This is perhaps more of a question, but I'm flagging it as a weakness because lack of clarity on this seems to undermine one of the main points -- that exploding/vanishing gradients are addressed. I.e., it seems that equal-gain guarantees are only guaranteed at the initialization? What evidence is provided for stability long-term?
- Not exactly a weakness either (to the extent to which it just describes an observation!) but the activations (esp Hermite) seem to add a high computational overhead.

**Questions:**

- Why do you choose Hermite, Fourier, and Tropic activations? Are there other possibilities? What main problems with existing methods to these overcome?
- Are there assumptions regarding distributions that aren't reasonable to expect, and to what extent do you need to check for them? E.g., are Gaussian or uniform distributions assumed by these activation functions, and would that be realistic?
- The Fourier is described in sine-cosine form, but the recommended initialization seems to only make use of a_k? Which coefficients are actually initialized and trained?

---

> ### Author Response · Authors · 2025-11-18
> **Answer to reviewer 3rgG [1/2]**
>
> We would like to sincerely thank reviewer 3rgG for their insightful comments and questions.
>
> ### Older datasets and models
> > It's not technically a weakness, but convention seems to be moving away from older datasets like CIFAR-10, and older models like GPT-2. This does not sway the decision much, but it seems to upset 'the community' and it does limit the generalizability of your claims, especially to larger scales.
>
> Thank you for your comment. We share the same concern, which is why we are actively working on adding/using larger models and newer benchmarks. Our preliminary results on 8B large language models trained on hundreds of billions of tokens, though not included in this paper, appear to confirm the trends reported here. We plan to present these results in a future study. Additionally, we included comparisons with older benchmarks and models because previous methods were based on them, and we have been previously asked to provide such comparisons.
>
> ### Does stability at initialization guarantee stability during training?
> > This is perhaps more of a question, but I'm flagging it as a weakness because lack of clarity on this seems to undermine one of the main points -- that exploding/vanishing gradients are addressed. I.e., it seems that equal-gain guarantees are only guaranteed at the initialization? What evidence is provided for stability long-term?
>
> This is an excellent question, and perhaps the most important one. However, before answering, we might ask the same for established classical activation such as ReLU, GELU, or SiLU. The Xavier Glorot or Kaiming He initializations usually used for those activations guarantee stability at initialization only. In practice, the trajectory of a training using gradient descent may still encounter exploding (NaN) or vanishing (0) values even with good initialization, which is why normalization methods such as BatchNorm or LayerNorm are widely used. The proposed initialization scheme we show in our work is simply a generalization of the variance-preserving initialization (proposed by He et al.) to the case of learnable polynomial, trigonometric, and tropical activations.
>
> In our work, we always assumed that both the input signal $x$ and its gradient $\Delta x$ follow a distribution of mean 0 and variance 1 (see Assumption 3.4). This is usually guaranteed by normalization layers. Typically, inputs are normalized at the start of training by subtracting the mean and dividing by the standard deviation of the data distribution. Features are further normalized dynamically at multiple stages of the network using BatchNorm, or LayerNorm ... Under these standard practices, assuming the input follows a standard normal or uniform distribution with mean 0, standard deviation 1, is sufficient. A promising future direction is to remove these normalization processes, and our preliminary experiments on large models in this direction, especially with high-degree Tropical activations, are encouraging, though beyond the scope of the current work.
>
> Empirically, in all experiments that we conducted, proper initialization was sufficient to carry training to convergence. In addition, in larger-scale experiments (models with billions of parameters and long token horizons, not reported here), initialization similarly prevented exploding or vanishing activations.
>
> ### Computational overhead
> > Not exactly a weakness either (to the extent to which it just describes an observation!) but the activations (esp Hermite) seem to add a high computational overhead.
>
> Thank you for pointing this out. It is true that some activations, particularly Hermite, introduce additional computational overhead compared to standard activations like GELU or SiLU. Table 11 in Appendix P (and Figures 19 and 20) quantifies per-epoch training times across different MLP widths and depths on an  NVIDIA A100 GPU.
> The overhead depends on several factors: the polynomial degree $d$, the network dimensions: depth and, to a lesser extent, width. For shallower networks, the slowdown is negligible (close to x1), while deeper networks exhibit higher overhead (up to x1.48 slowdown for Hermite at depth 32). Tropical and Fourier activations remain much closer to baseline (x1.05 – x1.19 slowdowns for most configurations). While current implementations are slower, the added FLOPs are comparable to GELU, and further optimization on dedicated hardware could significantly reduce this overhead. For example, recent work  (Raffel & Chen, 2025) has shown that memory bottlenecks for learnable rational activations can be mitigated, achieving speedups of up to x86.5 compared to unoptimized GPU kernel implementations of rational learnable activations in KAT (Yang & Wang, 2024).
>
> ---
> [Raffel & Chen, 2025]
> FlashKAT: Understanding and Addressing Performance Bottlenecks in the Kolmogorov-Arnold Transformer.
>
> [Yang & Wang, 2024] Yang, X., & Wang, X. Kolmogorov-Arnold Transformer. In The 13th International Conference on Learning Representations.

---

> ### Author Response · Authors · 2025-11-18
> **Answer to reviewer 3rgG [2/2]**
>
> ## Questions:
> ### Choice of Hermite, Fourier, and Tropical activations
> > Why do you choose Hermite, Fourier, and Tropic activations? Are there other possibilities? What main problems with existing methods to these overcome?
>
> We assumed normal and uniform input distributions, leading to Hermite and Fourier activations, respectively. Those are appropriate for these cases because they are the orthonormal bases associated with the scalar products derived from the probability measures of the normal and uniform distributions, respectively. Considering these bases is useful when computing moments, and the analytical integral computation is simplified by the orthogonality property. Furthermore, orthogonality ensures minimal coefficient count for the same degree of representation. In contrast, tropical polynomials are not bases of a Hilbert space, but they are orthogonal in the proximal sense. They were chosen as an example of a non-orthogonal basis that generalizes ReLU activations while capturing piecewise-linear behaviors.
>
> Many other bases could be used, including Legendre, Chebyshev, ... or other orthogonal polynomials that are members of what is known as the Askey scheme, as well as orthogonal wavelets (see our discussion), which can be constructed with orthogonal functions. Similarly, other piecewise-linear families such as tropical polynomials or rational functions could be considered. We chose these three classes as representative examples because they illustrate the main diverse functional spaces (orthogonal polynomial, orthogonal Fourier-type, and piecewise-linear) while keeping the study interpretable and computationally feasible.
>
>  ### Input distribution assumptions
> > Are there assumptions regarding distributions that aren't reasonable to expect, and to what extent do you need to check for them? E.g., are Gaussian or uniform distributions assumed by these activation functions, and would that be realistic?
>
> As explained before, normalization layers guarantee that the distribution of the input features is of mean 0 and variance 1 (see Assumption 3.4). However, normalization layers do not guarantee Gaussianity or Uniformity of the input distributions.
>
> More generally, inputs could follow arbitrary distributions, such as Gaussian mixtures, which can approximate any continuous distribution. The computation of the initialization gain has to be adapted. Still, due to the orthogonality, the moment calculation can be performed by hand, although it is more difficult to calculate analytically since a basis change must be performed for each Gaussian. Additionally, one can approximate this integral via numerical methods if calculation by hand becomes infeasible. This would have minimal computational overhead if done during the initialization phase.
>
> Overall, while Gaussian or uniform assumptions simplify the theory, they are not required in practice, and the framework can accommodate general input distributions with only a few adjustments at initialization.
>
>  ### Phase-amplitude form and sine-cosine form of Fourier activation
> > The Fourier is described in sine-cosine form, but the recommended initialization seems to only make use of a_k? Which coefficients are actually initialized and trained?
>
> In the sine-cosine form, the learnable coefficients are $a_k$: cosine amplitude, $b_k$: sine amplitude, and $f_k$: the frequencies, $k \in \mathbb{N}$.
> For the amplitude-phase form, the learnable coefficients are $A_k$: the amplitudes, $f_k$: the frequencies, and $\phi_k$: the phases.
> Both forms are mathematically equivalent, and we use an equivalent initialization as explained in line 275 **Alternative Fourier formula**. We also have the relationship: $A_k = \sqrt{a_k² +b_k²}$ and $\phi_k = \text{atan2}(b_k, a_k)$.
>
> The Fourier activation is described in sine-cosine form for the theoretical derivation of the initialization; in practice, we use the amplitude-phase form for implementation, as it requires fewer FLOPs. This is the version used throughout all experiments.

---

### Official Review · Reviewer_4XrB · 2025-10-30

**Soundness:** 3
**Presentation:** 3
**Contribution:** 3
**Rating:** 8
**Confidence:** 2

**Summary:**

The authors introduce a novel framework to enable learnable activation functions in deep neural networks. In particular, they focus on functions based on orthogonal bases and tropical polynomials. An initialization method for the activation functions  is introduced and the results showcase improvements over static functions. The efficacy of the method is benchmarked across vision and language tasks.

**Strengths:**

- The main idea is novel and well-motivated.

- The thorough theoretical support on the initialization methods is a valuable contribution to the community.

- I appreciate the benchmarking of the method across both text and vision tasks.

- The latency analysis is an important addition.

**Weaknesses:**

- I am missing an ablation over different backbones for both vision and language benchmarks.

- Although not a major weakness, additional experimental support on challenging benchmarks would increase the impact of the paper, e.g., on COCO for vision related tasks.

- A discussion on the application of the proposed activation functions for generative models (e.g., diffusion-based models) would be interesting.

Minor:

- The last sentence in ln. 485 seems to end abruptly.

**Questions:**

I would appreciate if the authors address the issues raised in the weaknesses section.

---

> ### Author Response · Authors · 2025-11-18
> **Answer to reviewer 4XrB**
>
> We sincerely thank the reviewer 4XrB for their detailed feedback and valuable suggestions.
>
> ### Backbone ablation
> > I am missing an ablation over different backbones for both vision and language benchmarks.
>
> Thank you for the suggestion. We conducted a preliminary study in which we tested the presented different activations on another vision backbone (ViT-tiny) using the same random seed. Similar trends to those reported for the ConvNeXt model were observed. For parsimony, we only performed the full experimental setup (with five different random seeds) for one convolutional neural network and one transformer: one for a computer vision task and the other for a natural language processing task.
> Evaluating additional backbones, such as ViT for CV or BERT for NLP, would strengthen the study, and we agree. Unfortunately, we currently lack the computational resources to run these ablations, but we plan to include them in future work or in a revised version of the paper if resources become available before December.
>
> ### More challenging benchmarks
> > Although not a major weakness, additional experimental support on challenging benchmarks would increase the impact of the paper, e.g., on COCO for vision related tasks.
>
> Indeed, we agree that evaluating on more challenging computer vision tasks will strengthen our paper. We plan to add a full object detection on COCO evaluation and/or an experiment on image segmentation in future work or in an updated version of the paper once computational resources permit it.
>
> ### Generative diffusion-based models discussion
> > A discussion on the application of the proposed activation functions for generative models (e.g., diffusion-based models) would be interesting.
>
> We agree that exploring the proposed activations within generative vision tasks would be an interesting approach. In the updated version of our submission, we will include a brief discussion of the potential applications and challenges of integrating our activations into diffusion and other computer vision generative models. Recent relevant works that motivate this approach include: Du et al. (2025), Liu & Tang (2025), Finder et al. (2024), Phung et al. (2023), and Qiu et al. (2023), but are not limited to these ... We would greatly appreciate any insights or suggestions from the reviewer regarding promising approaches, relevant literature, or potential challenges in integrating these activations into generative diffusion models.
>
> ---
> [Du et al., 2025]
> Du, Peng, et al. "Diffusion Transformer meets Multi-level Wavelet Spectrum for Single Image Super-Resolution." Proceedings of the IEEE/CVF International Conference on Computer Vision. 2025.
>
> [Liu & Tang, 2025]
> Liu, X., & Tang, H. (2025). DiffFNO: Diffusion Fourier Neural Operator. In Proceedings of the Computer Vision and Pattern Recognition Conference (pp. 150-160).
>
> [Finder et al., 2024]
> Finder, Shahaf E., et al. "Wavelet convolutions for large receptive fields." European Conference on Computer Vision. Cham: Springer Nature Switzerland, 2024.
>
> [Phung et al., 2023]
> Phung, Hao, Quan Dao, and Anh Tran. "Wavelet diffusion models are fast and scalable image generators." Proceedings of the IEEE/CVF conference on computer vision and pattern recognition. 2023.
>
> [Qiu et al, 2023]
> Qiu, Z., Liu, W., Feng, H., Xue, Y., Feng, Y., Liu, Z., ... & Schölkopf, B. (2023). Controlling text-to-image diffusion by orthogonal finetuning. Advances in Neural Information Processing Systems, 36, 79320-79362.
>
> ### Minor:
>
> > The last sentence in ln. 485 seems to end abruptly.
>
> For space considerations, we had to shorten our conclusion. We will expand this concluding sentence in the additional 10th page of the camera-ready version if the paper is accepted. Thanks.

---

### Official Review · Reviewer_CZi7 · 2025-11-01

**Soundness:** 4
**Presentation:** 4
**Contribution:** 3
**Rating:** 6
**Confidence:** 4

**Summary:**

This manuscript introduces a family of learnable activation functions based on orthogonal function bases (Hermite and Fourier) and tropical polynomials. The authors propose a variance-preserving initialization scheme to ensure stable gradient propagation and demonstrate the feasibility of using these activations in deep architectures such as ConvNeXt and GPT-2. The paper combines theoretical analysis, efficient implementations, and empirical validation, suggesting that polynomial activations can indeed yield competitive results with proper initialization.

**Strengths:**

1. The paper provides a rigorous variance-preserving initialization framework that unifies different activation families under an orthogonal function perspective. This is both mathematically elegant and practically meaningful.
2. By addressing Hermite, Fourier, and tropical bases, the study gives a broad view of orthogonal and piecewise-linear activations, including insightful links to classical activations (ReLU, GELU).
3. Experiments on ImageNet (ConvNeXt) and OpenWebText (GPT-2) convincingly demonstrate that the proposed activations can be trained stably and achieve comparable or slightly better performance than standard nonlinearities.
4. The inclusion of recursive formulations, efficient kernels, and open-sourced code (torchortho) greatly improves the work’s reproducibility and potential impact.

**Weaknesses:**

1. The reported 30–90% slower training speed (Section 6) is significant. The paper would benefit from more detailed timing analyses and GPU utilization comparisons to quantify the trade-off between performance and efficiency.
2. The experiments focus on classification and next-token prediction tasks. Additional ablations (e.g., fine-tuning, transfer learning, adversarial robustness) could help demonstrate broader applicability.

**Questions:**

1. The variance-preserving initialization ensures equal forward and backward gains for orthogonal activations. How sensitive is this balance to deviations from the assumed input distributions (e.g., non-Gaussian inputs during training)?
2. When replacing activations in large pretrained models (e.g., GPT-2), how does initialization interact with layer normalization and residual scaling? Are any stability adjustments required?
3. Have the authors explored methods to reduce computational cost, such as approximate polynomial evaluation (e.g., Chebyshev truncation, low-rank projection, or kernel-based approximation)? Could these reduce FLOPs while preserving stability?

---

> ### Author Response · Authors · 2025-11-19
> **Answer to reviewer CZi7 [1/2]**
>
> We respectfully thank reviewer CZi7 for the careful evaluation of our work. We provide our responses to the comments and questions below.
> ### Computational overhead
> > The reported 30–90% slower training speed (Section 6) is significant. The paper would benefit from more detailed timing analyses and GPU utilization comparisons to quantify the trade-off between performance and efficiency.
>
> We thank the reviewer for raising this point. The slowdown percentages mentioned in the comment do **not** appear in the current submitted version of our paper. They correspond to older preprint versions. We kindly invite the reviewer to refer instead to the version submitted to ICLR 2026, in which we rigorously benchmarked the timing of the introduced methods, and to disregard previous or preprint versions, which only provided rough estimates. However, we understand the concern regarding runtime overhead. In the present submission, we have added a detailed throughput benchmark in Appendix P, reporting per-epoch training times across multiple widths, depths and activation families. These results demonstrate that the overhead is minimal for shallow networks, becoming more noticeable with increasing depth. Notably, Tropical and Fourier activations remain close to the GELU baseline performance.
>
> Table 11 in Appendix P and Figures 19 and 20 quantify per-epoch training times on an NVIDIA A100 GPU across different MLP widths and depths. Overhead depends on several factors, including the polynomial degree and network dimensions (depth and, to a lesser extent, width). For shallower networks, the slowdown is negligible (close to x1), while deeper networks exhibit higher overhead (up to a x1.48 slowdown for Hermite at a depth of 32). Tropical and Fourier activations remain much closer to the baseline (slowdowns ranging from x1.05 to x1.19 for most configurations). Although current implementations are slower, the added FLOPs are comparable to those of GELU. Further optimization on dedicated hardware could significantly reduce this overhead. Recent work (Raffel & Chen, 2025) has shown, for example, that memory bottlenecks for learnable rational activations can be mitigated. This achieves speedups of up to x86.5 compared to unoptimized GPU kernel implementations of rational learnable activations in KAT (Yang & Wang, 2024).
>
> ---
> [Raffel & Chen, 2025] FlashKAT: Understanding and Addressing Performance Bottlenecks in the Kolmogorov-Arnold Transformer.
>
> [Yang & Wang, 2024] Yang, X., & Wang, X. Kolmogorov-Arnold Transformer. In The 13th International Conference on Learning Representations.
>
> ### Additional experiments
> > The experiments focus on classification and next-token prediction tasks. Additional ablations (e.g., fine-tuning, transfer learning, adversarial robustness) could help demonstrate broader applicability.
>
> Thank you for the suggestion. Fine-tuning results are already included in Section 4.5 and Appendix O, where we fine-tune a ConvNeXt model pretrained on ImageNet for image classification on CIFAR10. This experiment shows that the proposed learnable activations have a clear superiority in a fine-tuning setting.
> Additional tasks such as transfer learning or adversarial robustness are valuable directions, and we plan to explore them in future work or include them if time and compute permit.
>
> ## Questions
> ### Deviation from assumed input distributions
> > The variance-preserving initialization ensures equal forward and backward gains for orthogonal activations. How sensitive is this balance to deviations from the assumed input distributions (e.g., non-Gaussian inputs during training)?
>
> In practice, the initialization is reasonably robust to deviations, such as moderate non-Gaussianity, non-uniformity, or mild changes in kurtosis, because normalization layers maintain a zero mean and unitary variance of the activations throughout training.
>
> However, large deviations can break the equal-gain condition and lead to exploding or vanishing numerical values if no normalization is used. This is similar to classical activations, such as ReLU or GELU, whose variance-preserving initializations also rely on distributional assumptions.
>
> In all our experiments across different scales with standard normalization,  we did not observe instability, even when the inputs were noticeably non-Gaussian for Hermite activation and non-uniform for Fourier activation.

---

> ### Author Response · Authors · 2025-11-19
> **Answer to reviewer CZi7 [2/2]**
>
> > When replacing activations in large pretrained models (e.g., GPT-2), how does initialization interact with layer normalization and residual scaling? Are any stability adjustments required?
>
> Residual connections do not pose issues because they take place after the MLP layer. In subsequent blocks, a normalization layer is usually used before the following MLP, which guarantees the mean 0 and varaince 1 assumption (3.4). Normalization (LayerNorm, BatchNorm ...) remain important, just as with ReLU, GELU, or SiLU ... because variance-preserving initialization only guarantees stability at initialization. During training, normalization layers dynamically ensure that both activations and gradients remain approximately zero-mean and unit-variance, which is the assumption underlying our initialization (Assumption 3.4).
>
> In our work, we have always assumed that both the input signal $x$ and its gradient $\Delta x$ follow a distribution with a mean of 0 and a variance of 1. This is usually guaranteed by normalization layers. Typically, inputs are normalized at the start of training by subtracting the mean and dividing by the standard deviation of the data distribution. Features are further normalized dynamically at multiple stages of the network using BatchNorm, or LayerNorm ... Under these standard practices, assuming the input follows a standard normal or uniform distribution with mean 0, standard deviation 1, is sufficient. A promising future direction is to remove these normalization processes, and our preliminary experiments on large models in this direction, especially with high-degree Tropical activations, are encouraging, though beyond the scope of the current work.
>
> Empirically, in all experiments that we conducted, proper initialization was sufficient to carry training to convergence. In addition, in larger-scale experiments (models with billions of parameters and long token horizons, not reported here), initialization similarly prevented exploding or vanishing activations.
>
> > Have the authors explored methods to reduce computational cost, such as approximate polynomial evaluation (e.g., Chebyshev truncation, low-rank projection, or kernel-based approximation)? Could these reduce FLOPs while preserving stability?
>
> **Chebyshev truncation**: We are not certain  what reviewer CZi7 means by “Chebyshev truncation.” If it refers to considering an activation of the type: $\sum_{k=0}^{d} a_k T_k(x)$, with $d$ the degree of the truncation, $a_k$ the coefficients of the decomposition and $T_k$ the Chebyshev polynomials of degree $k$, then this is already similar to what we do with Hermite polynomials $He_k$ or Fourier series $sin(kx)$, $cos(kx)$ ... Remark 3.10 in our paper shows what assumptions must be made about the input distribution to handle the case with Chebyshev polynomials. Otherwise, the choice of a family of orthogonal polynomials (any family in the Askey scheme) is entirely feasible given that the suitable initialization is performed.
>
> **Low rank approximations**: Here again, we do not understand precisely what reviewer CZi7 means by “Low rank approximations”. If the reviewer refers to low rank approximation as in the Low Rank Adapter method for finetuning large models, then this approach is completely orthogonal to our activation methods and could be used in conjunction without conflict.
>
> If the reviewer refers to the notion of rank in polynomial theory, as in Warring rank, i.e, the least number of terms in a power-sum decomposition of a polynomial, then this notion could provide a promising direction to explore minimal representations of neural networks as polynomial mappings expressed in power-sum (product-sum) form. Some loose bounds on this minimal number of terms are given by Blekherman & Teitler (2015). This line of investigation is independent of our current work, but it could inspire future directions for designing minimal yet expressive polynomial mappings.
>
> **Kernel methods**: Kernel methods can also be considered, particularly Reproducing Kernel Hilbert Space (RKHS) methods. Activation could be defined as:
>
> $$F(x) := \sum_{i=1}^n \alpha_i K(x, c_i)$$
>
> with $\alpha_i$ and $c_i$ learnable, and $K$ a kernel (which can be a polynomial kernel, Gaussian kernel, or other...). Radial basis function (RBF) activations are a good example of this.
>
> Symmetric positive definite (SPD) kernels can also be considered, and by virtue of Mercer's theorem, they can easily be decomposed according to an orthonormal basis in the Hilbert space where we are working...
> Tropical reproducing kernels, a more sophisticated alternative,  can be considered, provided that we agree on the definition of the tropical kernel we want to use. See Aubin & Gaubert (2024).
>
> ---
> Blekherman, Grigoriy, and Zach Teitler. "On maximum, typical and generic ranks." Mathematische Annalen 362.3 (2015).
>
> Aubin, Pierre-Cyril, and Stéphane Gaubert. "Tropical reproducing kernels and optimization." Integral Equations and Operator Theory (2024).

---

### Official Review · Reviewer_MYHC · 2025-11-01

**Soundness:** 3
**Presentation:** 3
**Contribution:** 2
**Rating:** 4
**Confidence:** 4

**Summary:**

This paper explores the utilization of orthogonal polynomial activation functions in neural networks, specifically, first deriving the variance preserving initialization for Hermite, Fourier, and Tropical activation function, then conducting experiments on image classification and NLP tasks.

**Strengths:**

1. The paper is well written, presented with a clear structure.
2. The theorem-proof logic is clear and rigorious.
3. The visualization helps explain the conclusion.

**Weaknesses:**

Despite the strengths, here are some weaknesses:
1. The motivation is not clear, is it just an exploration on activations? And I am not sure if the proposed activation functions solve any existing problems. (Although I do know that not all innovative thought must solve something discrete, but I do suggest the author to refine this part.)
2. Since the paper is not the first to design a new kind of activation function, even not the first to use orthogonal  polynomials, I am not sure what is the core innovation.
3. The experiments show very little improvements when Hermite activation function is used. The tropical and Fourier activation function even have worse performance then GELU. These results restrict the value of the paper.
4. More experiments should be conducted, like, more tasks, more models, and more benchmarks.
5. In Proposition C.3, when computing the expectation of $F(x)^2$, the paper uses $\int F^2(x) \dfrac{e^{-x^2/2}}{\sqrt{2\pi}}dx$. I do not think adding the $\dfrac{e^{-x^2/2}}{\sqrt{2\pi}}$ term is rigorious, although it is based on the definition of Hermite. This problem is common for orthogonal polynomials, like Legendre, Hermite, and Chebyshev polynomials, that when adding the orthogonal term, the derivation is much easier.
6. Although the newly designed functions have negligible redundent parameters, it may cause low numerical stability. I am not sure how the authors resolve this problem.

**Questions:**

Please see the 'Weaknesses' section.

---

> ### Author Response · Authors · 2025-11-18
> **Answer to reviewer MYHC [1/2]**
>
> We would like to express our gratitude to reviewer MYHC for his insightful commentary. In the following, we will address the reviewer's concerns.
>
> ### Motivation
> > The motivation is not clear, is it just an exploration on activations? And I am not sure if the proposed activation functions solve any existing problems. (Although I do know that not all innovative thought must solve something discrete, but I do suggest the author to refine this part.)
>
>
> Thank you for the feedback. To clarify, our work is motivated by the question posed in the opening sentence of our abstract: Which functions can be used as activations in deep neural networks?
>
> Building on the Kolmogorov-Arnold Transformer work [Yang & Wang, 2024], which used learnable rational activations, we showed that other function families, such as polynomials (despite their unbounded nature), trigonometric polynomials, and the tropicalization of polynomials, can also effectively serve as activations in deep networks. This provides a general, structured overview of the space of possible nonlinear functions in deep neural networks.
>
> A key contribution of our work is our addressing of the instability of these activation functions via a variance-preserving initialization scheme inspired by Xavier Glorot and Kaiming He's initializations. This is a concrete problem that arises specifically in deep networks. As depth increases, the effective degree of these polynomial activations grows, causing numerical issues that do not appear in randomly initialized shallow networks. Empirically, our initialization method demonstrates the stability of training deep convolutional and transformer models with the proposed activations when trained on large-scale datasets.
>
> Furthermore, our work broadens our understanding of how these activation functions behave in deep models. It also offers a framework to interpret polynomially activated neural networks as multivariate polynomials, which helps demystify the nature of their function space.
>
> ---
> [Yang & Wang, 2024]
> Yang, X., & Wang, X. Kolmogorov-Arnold Transformer. In The Thirteenth International Conference on Learning Representations.
>
> ### Novelty
> > Since the paper is not the first to design a new kind of activation function, even not the first to use orthogonal polynomials, I am not sure what is the core innovation.
>
> The paper's innovation lies in its initialization method, which is based on a variance-preserving initialization, inspired by He and Xavier initializations. This initialization enabled us to achieve stability throughout training for **deep** networks using large datasets with no additional mechanisms. This initialization is necessary for deep networks because a random initialization leads to exploding (NaN) values or vanishing (0) values during training as the degree grows exponentially with the depth. Our article does not claim to be the first to use the presented activations. A glance at Section 2, "Related Work," and Appendix K, "Extended Related Work," reveals the extensive effort we made to provide a detailed history of these methods.
>
> ### Significance
> > The experiments show very little improvements when Hermite activation function is used. The tropical and Fourier activation function even have worse performance then GELU. These results restrict the value of the paper.
>
> We respectfully disagree that the improvements are "very little." While the differences may appear modest at first, they are statistically significant and consistent across runs, as evidenced by the p-values calculated across five different random seeds. In particular, Hermite activations reliably improve accuracy over GELU rather than fluctuating within noise. In Table 2, all three introduced activations performed better than GELU.

---

> ### Author Response · Authors · 2025-11-18
> **Answer to reviewer MYHC [2/2]**
>
> ### Number of experiments
> >More experiments should be conducted, like, more tasks, more models, and more benchmarks.
>
> We agree that additional experiments are always possible, but this remark applies to most methodological papers. Our goal here is to establish a general, theoretically grounded framework rather than to benchmark every model against every dataset exhaustively. The experiments already cover two demanding large-scale settings: ConvNeXt on ImageNet-1k and GPT-2 on OpenWebText, showing that the proposed activations scale reliably in both vision and language tasks.
> The experiments strike a very good combination of NLP and vision tasks, with fully convolutional neural networks and attention-based transformers, which cover the most widely used families of models and tasks in deep learning.
> Extending to more tasks is a natural next step, but we believe the current results sufficiently support the paper's main claims.
>
> ### Proof details
> >In Proposition C.3, when computing the expectation of $F(x)^2$, the paper uses $\int F^2(x) \dfrac{e^{-x^2/2}}{\sqrt{2\pi}}dx$. I do not think adding the $\dfrac{e^{-x^2/2}}{\sqrt{2\pi}}$ term is rigorious, although it is based on the definition of Hermite. This problem is common for orthogonal polynomials, like Legendre, Hermite, and Chebyshev polynomials, that when adding the orthogonal term, the derivation is much easier.
>
> The term $\dfrac{e^{-x^2/2}}{\sqrt{2\pi}}$ comes from the fact that we seek orthogonality in a Hilbert space defined by the following inner (scalar) product:
>
>
> $$
> \langle f, g \rangle := \int_{\mathbb{R}} f(x)g(x)\varphi(x) dx.
> $$
>
> where $\varphi(x) := \frac{e^{-x^2/2}}{\sqrt{2\pi}}$ is the standard normal probability density function.
>
> This is not an ad-hoc trick but the natural weight for the Hilbert space in which we seek orthogonality. Orthogonality of the probabilists’ Hermite polynomials $\mathrm{He}_n$ is defined with respect to this inner product.
>
> If the input follows a standard normal random variable $X \sim \mathcal{N}(0,1)$, as assumed in the proof, then the orthonormality simplifies the integral in the moment calculation. The Gaussian weight $\varphi(x)$ is therefore not optional as it is required when using the Hermite basis, which is orthogonal only in $L^2(\mathbb{R}, \varphi)$.
>
> Similarly, for the Fourier activation, we assumed that the input follows a uniform distribution $X \sim \mathcal{U}(-\pi,\pi)$ (see assumption 3.11), the orthogonality in this case takes place in $L^2(-\pi, \pi)$, with the inner product defined as:
>
> $$
> \langle f, g \rangle := \frac{1}{2\pi}\int_{-\pi}^{\pi} f(x)g(x)dx.
> $$
>
> If helpful, we can add a brief sentence to Proposition C.3 explicitly stating that the computation takes place in the Hilbert space $L^2(\mathbb{R}, \varphi)$, and that we consider functions $F$ that are square integrable with respect to the weight $\varphi$, i.e: $F \in L^2(\mathbb{R}, \varphi)$.
>
>
>
> ### Purpose: numerical stability
> > Although the newly designed functions have negligible redundent parameters, it may cause low numerical stability. I am not sure how the authors resolve this problem.
>
> The entire purpose of our initialization scheme, and thus our paper, is precisely to ensure numerical stability for these learnable activations at initialization. The variance-preserving gain we derive guarantees that both activations and their gradients remain well-scaled at initialization, preventing exploding or vanishing behavior, even for high-degree polynomial, tropical, or Fourier activations. Empirically, all our large-scale experiments confirm that training remains stable without requiring any auxiliary mechanisms.

---

### Meta-Review · Area_Chair_SEsw · 2026-01-07

**Summary:**

The paper focuses on a simple question: what type of activation functions can be used in modern architectures?
The paper studies a family of learnable activation functions derived from orthogonal Hermite, Fourier, and tropical bases, introducing a variance-preserving initialization strategy designed to stabilize signal propagation in modern architectures. Although experiments on ConvNeXt and GPT-2 demonstrate trainability, the submission could improve the justification on the substantial computational overhead incurred.

**Reviewer Concerns:**

The reviewers raised various concerns from the experimental results and setup to the motivation of this work. Many of those concerns are valid and were originally not addressed in the paper, however, I find the answers for some of these concerns to be sufficient. Concretley, the main concerns of the reviewers are the following:

1. The benchmarks used for images are quite simple and not the most updated; experiments on larger datasets could be included.
2. Questions on the numerical stability of the activation functions.
3. Speed issues with the proposed activations.
4. What the motivation is for the activation functions and the proposed study.

**Reviewer Scores:**

I believe, the rebuttal addresses question 2 on the stability, and question 3 on the speedup, while the rebuttal also provides answers for the rest of the questions, but it's unclear whether it would have satisfied the reviewers. Question 4 is probably the most important concern raised by multiple reviewers and normally I would expect a longer discussion here. Even though this is not covered in the discussion, I believe that this question is justified by more recent papers that cover polynomials and other bases and are not cited at the moment (the camera-ready can fix this). This is also complemented by recent theoretical results on identifiability of polynomial bases that can offer additional arguments for using such functions. What I would strongly suggest to the authors is to broaden their related work in the camera-ready version, which would answer some of the questions of the reviewers and include many of the approaches mentioned above. Activation functions are not a new topic and there has been a lot of work also on activation function search (such as with NAS), including results on their initialization. Those results do not cover the proposed results, but they are complementary, so a comparison with those would be recommended.

---

### Decision · Program_Chairs · 2026-01-26

Accept (Poster)